# AAV2/9-mediated silencing of PMP22 prevents the development of pathological features in a rat model of Charcot-Marie-Tooth disease 1 A

Benoit Gautier [1✉], Helene Hajjar [1,10], Sylvia Soares [2], Jade Berthelot[1], Marie Deck [1], Scarlette Abbou [1], Graham Campbell[1], Maria Ceprian[1,11], Sergio Gonzalez[1,11], Claire-Maëlle Fovet [3], Vlad Schütza[4], Antoine Jouvenel[1], Cyril Rivat[1], Michel Zerah[5], Virginie François[6], Caroline Le Guiner[6], Patrick Aubourg[7,8], Robert Fledrich [4✉] & Nicolas Tricaud [1,9,11✉]

Charcot-Marie-Tooth disease 1 A (CMT1A) results from a duplication of the *PMP22* gene in Schwann cells and a deficit of myelination in peripheral nerves. Patients with CMT1A have reduced nerve conduction velocity, muscle wasting, hand and foot deformations and foot drop walking. Here, we evaluate the safety and efficacy of recombinant adeno-associated viral vector serotype 9 (AAV2/9) expressing GFP and shRNAs targeting *Pmp22* mRNA in animal models of Charcot-Marie-Tooth disease 1 A. Intra-nerve delivery of AAV2/9 in the sciatic nerve allowed widespread transgene expression in resident myelinating Schwann cells in mice, rats and non-human primates. A bilateral treatment restore expression levels of PMP22 comparable to wild-type conditions, resulting in increased myelination and prevention of motor and sensory impairments over a twelve-months period in a rat model of CMT1A. We observed limited off-target transduction and immune response using the intra-nerve delivery route. A combination of previously characterized human skin biomarkers is able to discriminate between treated and untreated animals, indicating their potential use as part of outcome measures.

[1] INM, Univ. Montpellier, INSERM, Montpellier, France. [2] Sorbonne Université, CNRS, INSERM, IBPS, Neuroscience Paris Seine, Paris, France. [3] INSERM U1184, Immunology of Viral, Auto-immune, Hematological and Bacterial Diseases (ImVA-HB), IDMIT Department, CEA, Fontenay-Aux-Roses, France. [4] Institute of Anatomy, Leipzig University, Leipzig, Germany. [5] Paediatric Neurosurgery Department, Université Paris Descartes and Assistance Publique-Hôpitaux de Paris, Hôpital Universitaire Necker, Paris, France. [6] INSERM UMR 1089, Université de Nantes, CHU de Nantes, Nantes, France. [7] Department of Paediatric Neurology, Centre Hospitalier Universitaire de Bicêtre, Le Kremlin-Bicètre, France. [8] INSERM U1169, Université Paris-Sud, Orsay, France. [9] I-Stem, UEVE/UPS U861, INSERM U861, AFM, Corbeil-Essonnes, France. [10]Present address: Institute of Regenerative Medicine and Biotherapies (IRMB), University of Montpellier, INSERM, Montpellier, France. [11]Present address: Laboratory of Pathogen-Host Interactions (LPHI), University of Montpellier, Montpellier, France. ✉email: benoit.gautier@inserm.fr; robert.fledrich@medizin.uni-leipzig.de; nicolas.tricaud@inserm.fr

Peripheral nerves that bundle axons emanating from neuronal cell bodies are found throughout the body forming the peripheral nervous system (PNS). These axons are covered with Schwann cells and those of large caliber are wrapped in a myelin sheath made by myelinating Schwann cells (mSC) to assure an extremely fast nerve conduction (up to 100 m/s). The myelin forms several successive segments named internodes, which electrically isolate the axonal membrane except at unmyelinated nodes of Ranvier where action potentials are propagated[1].

This myelin sheath is critical for both motor and sensory functions in humans. Indeed, several peripheral nerve diseases impair the myelin sheath leading to reduced nerve conduction velocity, nerve dysfunction, muscle wasting, limb extremities deformations and walking and sensory problems[2]. The large majority of patients suffering from hereditary diseases of peripheral nerves, namely Charcot-Marie-Tooth diseases (CMT), have defects in the myelin sheath formation, function or maintenance[3]. The most common of these myelin-related CMT diseases is CMT1A (prevalence: 0.5–1.5/10,000)[4,5]. This disease, resulting from the duplication of the *PMP22* gene, is characterized by a set of heterogeneous symptoms, the most serious being feet and hands deformation and walking impairment[4]. PMP22 protein is a transmembrane glycoprotein located in the myelin sheath. In CMT1A, PMP22 protein overexpression in Schwann cells leads to defects in the number of myelinated segments that are formed, short internodes, myelin sheath defects, myelin degeneration (demyelination) and finally axonal loss[6].

No cure exists for this disease, however, pharmacological treatments have been recently investigated[4]. Some preclinical studies in rodents demonstrated that antisense oligonucleotides targeting *Pmp22* mRNA expression significantly improved the phenotype[7–9], suggesting *PMP22* silencing is an effective way to tackle the disease. Feeding a phospholipid enriched diet to a rat model of CMT1A increased myelination, prevented axonal loss and dramatically delayed the occurrence of the disease[10]. However, the benefit only lasted as long as the treatment was administrated.

Gene therapy therefore constitutes an alternative therapy allowing for long term benefits for patients[11]. Proofs of concept for gene therapy using lentiviral vectors injected in the spinal cord of myelin-related CMT mouse models have been reported[12–14]. However, currently, in vivo gene therapy assays mostly use adeno-associated virus (AAV)-based strategies as these vectors do not integrate the genome of transduced cells, spread more easily in the tissues and display a limited immunogenicity[15]. AAV serotypes 2/9 and 2/rh10 are commonly used to transduce the central nervous system and several clinical trials are ongoing with promising results[11,16]. Moreover, an AAV2/9-based therapy recently obtained market authorization from FDA to treat spinal motor atrophy in infants[17]. Gene therapy may therefore represent an efficient and safe way to treat CMT1A in the long term. However, the pattern of transduction of Schwann cells with AAV2/9 or rh10 remains unclear.

Recent clinical trials for peripheral neuropathies, and in particular for CMT1A, have shown that the chronicity of the disease makes the evaluation of the treatment outcome very difficult. Composite scores have been shown to be the most reliable outcome measure, but they remain poorly discriminant[18–20]. Molecular biomarkers have been characterized for CMT1A and some of them allow discrimination on the basis of disease severity[21]. However, none of them has been validated yet as outcome measure for a therapy.

Here we show that mSC of mouse, rat and non-human primate sciatic nerves are widely and specifically transduced by AAV2/9 when injected directly into the nerve. Using an AAV2/9 expressing a small hairpin inhibitory RNA (shRNA) directed against *Pmp22* mRNA, we treated a rat model of CMT1A when myelination begins. A single injection treatment in both sciatic nerves prevented the disease symptoms for at least one year. The combined expressions of human biomarkers in the paw skin of rats correlated with the phenotype and allowed discrimination between treated animals and their sham-treated littermates, indicating that these markers can be used as an outcome measure of the treatment. In addition, the dispersion of the vector remained limited to the injected nerves and the humoral immune response generated against the vector remained barely detectable in injected animals. Taken together, this work suggests that an intra-nerve AAV2/9-mediated gene therapy represents an effective and attractive therapy for myelin-related CMT diseases.

## Results

**Broad and specific transduction of mSC is reached after AAV2/9 injection in sciatic nerves of mammals.** As the literature reported a limited transduction efficiency with several AAV serotypes[22,23], we investigated the transduction efficiency of recombinant single-stranded AAV2/9 and AAV2/rh10 vectors expressing enhanced GFP under a CAG promoter after injection into the sciatic nerves of mice and rats. These injections were done using a non-traumatic microinjection protocol previously described[24]. Briefly, fine glass needles containing a viral solution stained with Fast Green were introduced in the sciatic nerve and the solution was slowly injected using multiple short-time pressure pulses[24]. In these conditions, we observed a high amount of transduced GFP-expressing cells along the nerve of both species (Fig. 1a and Supplementary Fig. S1). To analyze the nature of these cells, we first imaged injected sciatic nerves using Coherent Antistokes Raman Scattering (CARS) non-linear microscopy technique[25]. This allows myelin sheath imaging without any labeling[25]. Using this technique on intact nerves in a longitudinal view, we observed that the majority of transduced cells were myelinated cells (Fig. 1b) and therefore mSC. In addition, typical morphological characteristics of mSC were seen through GFP labeling and/or subcellular markers: Schmidt-Lanterman incisures (Supplementary Fig. S2a and b, white arrows), Cajal's bands (Supplementary Fig. S2a, blue arrows) and paranodal loops surrounding the node of Ranvier (Supplementary Fig. S2a and c, arrowheads). To go further, sciatic nerve cross-sections were immunostained for axonal (Tuj1) and myelin (MBP) markers allowing to definitively identify mSC as GFP-labeled MBP-positive cells surrounding axons (Fig. 1c). As some non-myelinated cells (nmc) located in the nerve, such as non-myelinating Schwann cells and fibroblasts, are not easily seen in cross-sections, we teased sciatic nerve fibers on a glass slide and identified GFP-positive mSC, axons and nmc using their morphology and/or specific markers (Supplementary Fig. S3). On 100 cells transduced with AAV2/9-CAG-GFP vector after intra-nerve injection in rat pup, 95 were mSC, 4 nmc and 1 was an axon (Table 1). The same specificity was observed after intra-nerve injection in adult and pup rat and mouse for AAV2/9-CAG-GFP and in adult mouse and rat pup for AAV2/rh10-CAG-GFP (Table 1). Taken together the data highlight the specificity of AAV2/9 and AAV2/rh10 for mSC after intra-nerve injection.

Next, as each large axon is surrounded by mSC in nerves, we used nerve cross-sections immunostained for MBP and Tuj1 to measure the number of transduced mSC (GFP and MBP positive) surrounding axons (Tuj1 positive), as a measure of the transduction rate of mSC. Injections of $5 \times 10^{10}$ vg (vector genome)/nerve and $1.8 \times 10^{11}$ vg/nerve of AAV2/9-CAG-GFP in adult mouse and rat resulted in the transduction of 93% and 80% of mSC respectively at the injection site (Table 2). Similar

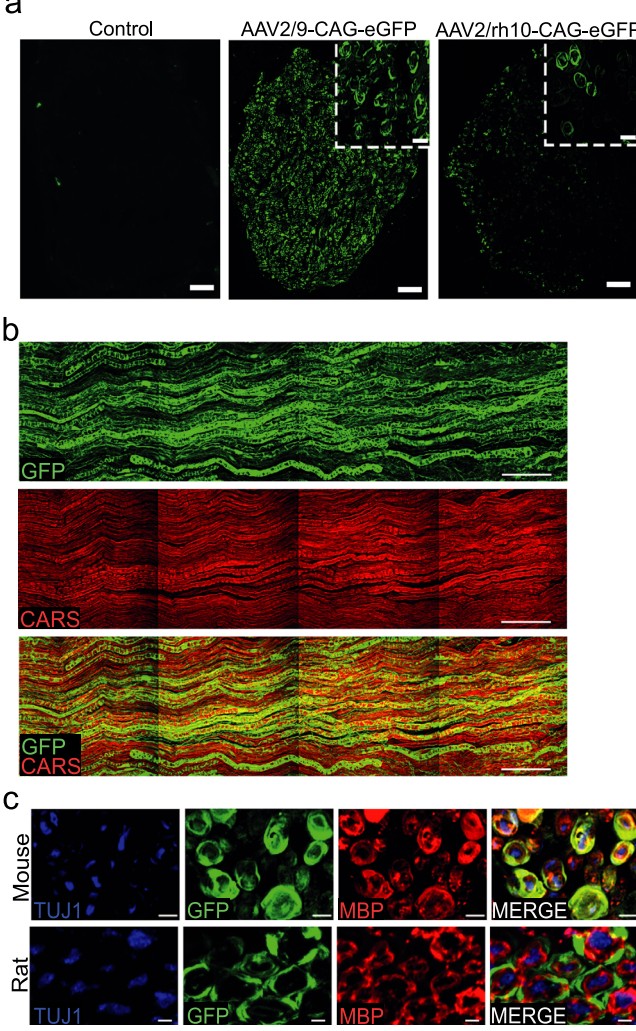

**Fig. 1 AAV2/9 and AAV2/rh10-CAG-GFP transduce mSC after intra-nerve injection in mouse and rat. a** Representative images of sciatic nerve cross-sections showing GFP protein expression at the injection site after intra-nerve injection of AAV2/9 and AAV2/rh10-CAG-GFP in rat pup (P6-P7, $1 \times 10^{11}$ vg/nerve in 8 μl, $n = 3$ animals per group). Control represents a rat pup sciatic nerve injected with Fast Green alone. All animals were sacrificed one month post-injection. Inserts show the circular shape of transduced cells. Scale bars: 50 μm and 10 μm for the inserts. **b** GFP (green) and CARS (red) imaging largely overlap in a sciatic nerve injected with AAV2/9-CAG-GFP in rat pup (P6-P7, $1 \times 10^{11}$ vg/nerve in 8 μl, $n = 3$ animals) twelve months post-injection. Scale bar: 100 μm. **c** Mouse and rat sciatic nerves cross-sections immunostained for myelin MBP (red) and axonal Tuj1 (blue) after AAV2/9-CAG-GFP injection (adult mouse injected at 2–3 months old with $5 \times 10^{10}$ vg/nerve in 8 μl, rat pup injected at P6-P7 with $1 \times 10^{11}$ vg/nerve in 8 μl, $n = 3$ animals for each species). GFP (green) partially colocalizes with MBP and both surround axons. Animals were sacrificed one month post-injection. Scale bar: 4 μm (mouse) and 2 μm (rat).

transduction efficiencies were also obtained after intra-nerve injection of AAV2/9-CAG-GFP in newborn mouse and rat nerves when myelination starts (Table 2). The injection of the same amounts of AAV2/rh10-CAG-GFP vector in adult mouse nerves and newborn rat nerves resulted in a lower transduction efficiency (Table 2).

In order to document the vectors ability to transduce primate mSC in vivo, we used a regular 22 G syringe to inject $5 \times 10^{12}$ vg/ nerve of each vector into the sciatic nerve of an adult non-human primate (NHP). As described in rodents, immunostaining of nerve cross-sections indicated that both vector transduced mSC in these conditions (Supplementary Fig. S4). Calculating the transduction rate as described previously, a high transduction rate for mSC was observed using AAV2/9-CAG-GFP but not using AAV2/rh10-CAG-GFP (Supplementary Fig. S4, Supplementary Table S1).

We next evaluated the diffusion of the vectors along mouse and rat sciatic nerves collecting injected nerve samples proximally and distally from the injection site in order to cover the full length of the nerve (Supplementary Fig. S5). This diffusion was very significant for AAV2/9-CAG-GFP as the average transduction rate was 73% at these two distant points in both mice and rats (Table 2). In the NHP sciatic nerve injected with AAV2/9-CAG-GFP vector, the diffusion was evaluated at three distant points distally and proximally of the injection point covering 30 to 50% of the nerve length. The transduction rate was 68% at the injection site, 69% 2 cm distally, 55% 2 cm proximally and 21% 4 cm proximally (Supplementary Table S1).

**Designing and validating a small hairpin inhibitory RNA to downregulate *Pmp22* expression in CMT1A rat model.** The high transduction rate of mSC in mouse, rat and NHP by AAV2/

---

**Table 1 Specificity of the transduction pattern after intra-nerve injection of AAV2/9 and AAV2/rh10-CAG-GFP in rodents.**

| | Cellular specificity of the transduction pattern after intra-nerve injection (%) | | | | | |
| --- | --- | --- | --- | --- | --- | --- |
| | AAV2/9-CAG-GFP | | | | AAV2/rh10-CAG-GFP | |
| | Mouse | | Rat | | Mouse | Rat |
| | Adult | Pup | Adult | Pup | Adult | Pup |
| **mSC** | 97 ± 3 | 87 ± 12 | 89 ± 9 | 95 ± 2 | 82 ± 17 | 95 ± 1 |
| **nmc** | 3 ± 3 | 2 ± 2 | 7 ± 5 | 4 ± 1 | 4 ± 4 | 4 ± 1 |
| **Axons** | 0 | 11 ± 11 | 4 ± 4 | 1 ± 1 | 14 ± 13 | 1 ± 1 |

Animal groups were composed of three animals. Mouse and rat pups were injected at P2-P3 and P6-P7 respectively. Adult mice and rats were injected at 2-3 months old. All animals were sacrificed one month post-injection. Sciatic nerves were then teased and analyzed. The specificity is the ratio of mSC (GFP and MBP positive cells), nmc (GFP and glial fibrillary acidic protein, GFAP, positive cells) or axons (GFP and Neurofilament, NF, positive cells) transduced on the overall number of transduced cells (GFP positive cells). The results are expressed as the mean ± SD. Source data are provided as a Source Data file.

---

**Table 2 Quantification of the transduction pattern after intra-nerve injection of AAV2/9 and AAV2/rh10-CAG-GFP in rodents.**

| | Transduction rate after intra-nerve injection (%) | | | | | |
| --- | --- | --- | --- | --- | --- | --- |
| | AAV2/9-CAG-GFP | | | | AAV2/rh10-CAG-GFP | |
| | Mouse | | Rat | | Mouse | Rat |
| | Adult | Pup | Adult | Pup | Adult | Pup |
| **Proximally** | 63 ± 24 | 74 ± 7 | NA | 81 ± 7 | 42 ± 22 | 53 ± 3 |
| **Injection site** | 93 ± 2 | 85 ± 15 | 80 ± 14 | 87 ± 1 | 51 ± 11 | 46 ± 5 |
| **Distally** | 91 ± 2 | 74 ± 14 | NA | 50 ± 24 | 42 ± 16 | 5 ± 6 |

Animal groups were composed of three animals. Mouse and rat pups were injected at P2-P3 and P6-P7 respectively. Adult mice and rats were injected at 2-3 months old. All animals were sacrificed one month post-injection. The transduction rate is the percentage of transduced mSC (GFP and MBP positive cells surrounding Tuj1 positive axons) on the overall number of mSC (MBP positive cells surrounding Tuj1 positive axons) per section. Proximal distances of the injection site were 2 cm for mice and 3 cm for rats. Distal distances of the injection site were 0.5 cm for mice and 1 cm for rats. The results are expressed as the mean ± SD. NA, Not available (not done). Source data are provided as a Source Data file.

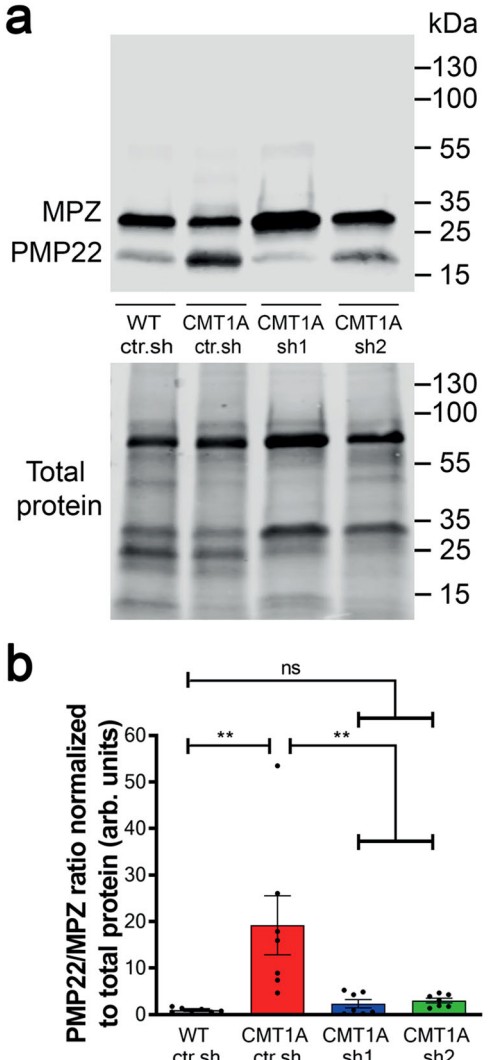

**Fig. 2 Intra-nerve injections of AAV2/9-sh1 and sh2 prevent PMP22 overexpression in CMT1A sciatic nerves. a** Representative image of Western blot showing PMP22 and MPZ protein levels (upper panel) and total protein as loading control (lower panel) in rat sciatic nerve lysates from WT ctr.sh, CMT1A ctr.sh, CMT1A sh1 or CMT1A sh2 three months after injection. **b** Graph shows the average level of PMP22 / MPZ ratio in sciatic nerve lysates normalized to total protein loaded as loading control ($n = 7$ animals per group). Statistical test shows one-way ANOVA followed by Tukey's post hoc, two-sided. **$p = 0.0027$ between WT ctr.sh and CMT1A ctr.sh, **$p = 0.0056$ between CMT1A ctr.sh and CMT1A sh1, **$p = 0.008$ between CMT1A ctr.sh and CMT1A sh2; ns, not significant; arb. units, arbitrary unit. All error bars show SEM. Source data are provided as a Source Data file.

9 prompted us to evaluate its use as a vector to carry a therapeutic tool into defective mSC in CMT1A disease. As CMT1A results from PMP22 overexpression, we looked for shRNAs targeting human *PMP22* mRNA in order to decrease PMP22 expression in CMT1A mSC. Two independent shRNAs or a control shRNA with no target were cloned in a pAAV plasmid under a U6 promoter followed by a CMV-GFP reporter cassette. Both shRNAs were found to be effective in reducing human PMP22 in HEK293 cells (Supplementary Fig. S6a), showing that decreasing PMP22 expression in mSC may represent a relevant therapeutic approach for CMT1A disease.

To go further and perform a proof of concept, we chose the rat model of CMT1A in which mouse *Pmp22* mRNA is

overexpressed[26]. Indeed, this model mimics the clinical aspects of the human disease more closely than any others: immediately after peripheral nerve myelination begins, nerves show hypomyelination, shorter internodal lengths, deficit in large fibers, hypermyelination of small fibers, myelin sheath defects and demyelination[27–29]. Two independent shRNAs (sh1 and sh2), cloned in pAAV as described previously, were found to significantly decrease mouse PMP22 protein level in a mouse Schwann cell line in a dose-dependent manner (Supplementary Fig. S6b). Only sh1 was effective on rat PMP22 level (Supplementary Fig. S6c) and both were ineffective to downregulate human PMP22 level (Supplementary Fig. S6a).

AAV2/9 vectors expressing sh1 or sh2 (AAV2/9-sh1 and AAV2/9-sh2) were then manufactured and injected bilaterally ($1 \times 10^{11}$ vg/nerve) in the sciatic nerves of CMT1A rat pups (P6-P7) (groups CMT1A sh1 and CMT1A sh2 respectively). WT littermates and CMT1A rat pups injected with AAV2/9 carrying a control shRNA (AAV2/9-ctr.sh) were used as controls (groups WT ctr.sh and CMT1A ctr.sh respectively). Treated and control animals were followed for as long as twelve months after injection.

We first investigated the efficiency of both AAV2/9-sh1 and AAV2/9-sh2 vectors to reduce PMP22 expression in mSC in vivo. At the mRNA level, mouse but not rat *Pmp22* mRNA was upregulated relative to myelin marker *Mpz* in control CMT1A rats, resulting in an overall higher *Pmp22* mRNA expression (Supplementary Fig. S7). However, neither AAV2/9-sh1 nor AAV2/9-sh2 treatment downregulated mouse or rat *Pmp22* mRNAs expression (Supplementary Fig. S7). At the protein level, Western blot analysis showed that PMP22 level normalized on the myelin marker MPZ was increased in CMT1A ctr.sh compared to WT ctr.sh (Fig. 2a, b). The transduction of mSC by AAV2/9-sh1 and AAV2/9-sh2 decreased PMP22 level back to control WT ctr.sh (Fig. 2a, b). No downregulation beyond that of control levels was observed in treated CMT1A animals. Sciatic nerve sections immunostained for PMP22 also showed a decreased PMP22 protein expression in mSC of treated animals (Supplementary Fig. S8). Taken together these data showed that AAV2/9 vectors carrying shRNAs targeting *Pmp22* were able to prevent PMP22 overexpression in mSC of CMT1A rats through a mechanism independent of the mRNA stability.

**AAV2/9-sh1 and -sh2 reduce myelinated fiber defects in the sciatic nerve of CMT1A rats.** As CMT1A disease affects mSC number and therefore myelin amount in nerves, we used Western blot to measure the amount of myelin marker MPZ over the total amount of protein after AAV2/9-sh1 and -sh2 treatment (Fig. 3a). While CMT1A rats showed a lower level of MPZ than WT ctr.sh rats, MPZ levels in CMT1A sh1 and sh2 rats were not statistically different from WT ctr.sh rats (Fig. 3b).

Second, we focused on the specific myelin sheath defects observed in the CMT1A rat model using CARS imaging on freshly fixed nerves. As described in detail previously[25], the CMT1A myelin sheath showed a high heterogeneity compared to WT rat littermates: thin myelin sheath or demyelination (Fig. 4a, white arrows), focal hypermyelination (Fig. 4a, blue arrows), and myelin degeneration with ovoids formation (Fig. 4a, stars). While still present, all these defective features were less abundant in CMT1A rats treated with AAV2/9-sh1 or –sh2 (Fig. 4a). A typical feature of CMT1A nerves is the shorter internodal distance[29,30]. Using CARS, we found that the number of nodes of Ranvier (Fig. 4a, b, arrowheads) per fiber increased in CMT1A ctr.sh nerves compared to WT ctr.sh nerves (Fig. 4b, c), indicating shorter internodes. When CMT1A rats were treated with AAV2/9-sh1 or -sh2, the number of nodes of Ranvier per fiber decreased significantly, showing that internodes were longer (Fig. 4c).

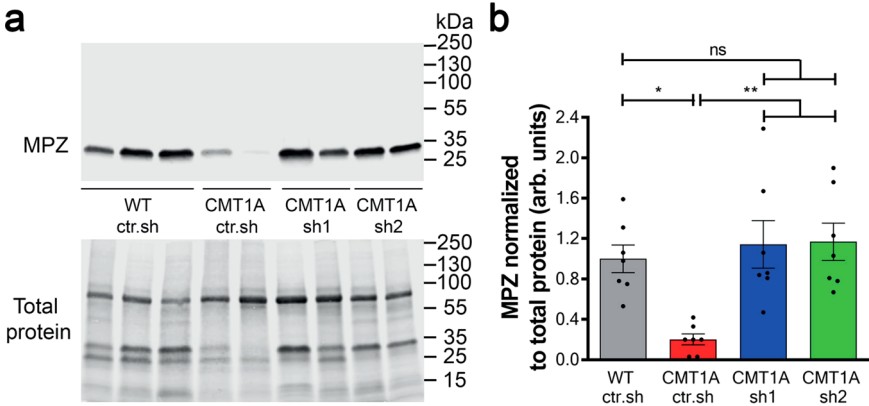

**Fig. 3 Intra-nerve injections of AAV2/9-sh1 and -sh2 in CMT1A rats normalize MPZ expression back to WT rat level. a** Representative images of Western blot showing MPZ protein levels (upper panel) and total protein as loading control (lower panel) in sciatic nerve lysates from WT ctr.sh, CMT1A ctr.sh, CMT1A sh1 or CMT1A sh2 three months after injection. **b** Graph shows mean MPZ level in sciatic nerve lysates normalized to total protein level loaded as loading control ($n = 7$ animals per group). Statistical test shows one-way ANOVA followed by Tukey's post hoc, two-sided. $*p = 0.0122$ between WT ctr.sh and CMT1A ctr.sh, $**p = 0.0028$ between CMT1A ctr.sh and CMT1A sh1, $**p = 0.0022$ between CMT1A ctr.sh and CMT1A sh2; ns, not significant; arb. units, arbitrary unit. All error bars show SEM. Source data are provided as a Source Data file.

Finally, we investigated the myelin sheath structure using electron microscopy semi-thin nerve sections. Light microscopy imaging of these sections showed typical features of rat CMT1A disease: lower density of myelinated fibers, hypomyelination (Fig. 4d, green arrows) or demyelination of large axons (Fig. 4d, orange stars) and hypermyelination of small axons (Fig. 4d, blue stars). These features were still present in the nerve of treated animals but at lesser extent (Fig. 4d). We found that AAV2/9-sh1 and -sh2 treatment significantly increased the myelinated fibers density (Fig. 4e), the number of large myelinated axons and proportionally reduced the number of small myelinated axons (Fig. 4f). Finally, following treatment, the g-ratio (axon diameter/ myelinated fiber diameter) was increased for small caliber axons and decreased for large caliber axons (Fig. 4g), which corrected values toward control WT ctr.sh values. Taking together, these data indicate that the treatment with AAV2/9-sh1 or -sh2 prevents myelin loss and the occurrence of myelinated fiber defects in CMT1A rats.

**AAV2/9-sh1 and -sh2 treatments prevent motor and sensitive defects on the long term in CMT1A rats.** As CMT1A is a myelin-related disease, one of the first symptoms to occur is the decrease of the nerve conduction velocity (NCV). Accordingly, NCV was significantly reduced in CMT1A ctr.sh animals as soon as one month after birth compared to control WT ctr.sh animals (Fig. 5a). When CMT1A animals were treated with AAV2/9-sh1 or-sh2, the NCV remained close to WT ctr.sh values at all-time points for at least twelve months (Fig. 5a).

The strong decrease of NCV in diseased animals correlated with a clumsy behavior when animals crossed a narrow beam and the treatment prevented this defect (Supplementary Videos 1 and 2). Motor behavior tests of rotarod and grip strength also showed reduced performances of CMT1A ctr.sh animals compared to control WT ctr.sh animals, starting two months after birth and these deficiencies were largely prevented by the treatment with AAV2/9-sh1 and –sh2 (Fig. 5b, c). No significant evidence of a reduction in the effectiveness of the treatment could be observed even twelve months after the treatment.

CMT1A disease shows a reduced sensory perception due to both a defect in myelinated sensory fibers and a loss of sensory axons on the long term[31]. Consequently, we performed Randall-Selitto test to assess the effects of the therapeutic vectors on the mechanical pain sensitivity of lower limbs at six months and

twelve months post-injection (Fig. 5d). We first observed an increase in the mechanical nociceptive threshold of CMT1A ctr. sh animals as compared to the WT ctr.sh group indicating hypoalgesia. Treatment with both AAV2/9-sh1 and –sh2 completely prevented this sensory deficit (Fig. 5d). Consequently, treatment with AAV2/9 vectors that reduces PMP22 levels in mSC constitutes an efficient and long-term preventive treatment for CMT1A symptoms in rats.

**Human skin biomarkers reliably and robustly discriminate treated from sham and wild-type animals.** Recent clinical trials for drugs targeting CMT diseases have illustrated the lack of outcome measures sensitive enough for such chronic peripheral neuropathies[32,33]. We therefore tested whether recently discovered human skin mRNA biomarkers[21] could be used as an outcome measure for our gene therapy efficiency. Forepaw glabrous skin was collected at 12 months in all cohorts and the expression of nine identified biomarkers was quantified using RT-qPCR. Some of these biomarker expressions were significantly differentially expressed in the skin of CMT1A sh1 and CMT1A sh2 animals versus CMT1A ctr.sh animals (Supplementary Fig. S9). However, the expression levels remained highly heterogeneous among animals, reflecting the variability of CMT1A disease expression that is recapitulated in transgenic rats[28]. We therefore performed a multi-variated principle component analysis (PCA), including the entire battery of nine validated human biomarkers. In this analysis, biomarkers allowed a significant segregation of WT ctr.sh animals from CMT1A ctr.sh animals, and of CMT1A sh1 and CMT1A sh2 animals from CMT1A ctr.sh animals (Fig. 6a). These data indicated that biomarkers expression in the paw skin allows for detecting the outcome of the gene therapy treatment in CMT1A rats.

To go further, we performed a correlation analysis including both functional phenotypes (Rotarod, grip strength test, Nerve Conduction Velocity and Randall-Selitto test) and biomarkers expression at 12 months. While correlations were more significant (stars and large disk size) within the phenotypic test group or more numerous within the biomarkers group (11/27), significant correlations occurred between functional phenotypes and biomarkers groups (10/27) (Fig. 6b). *Nrg1.1* (3 correlations), *Gria1* (2), *Cda* (2), *Gstt2* (1) *Anpep* (1) and *Enpp1* (1) were the genes whose skin expression correlated the best with the phenotype improvement of CMT1A rats following gene therapy.

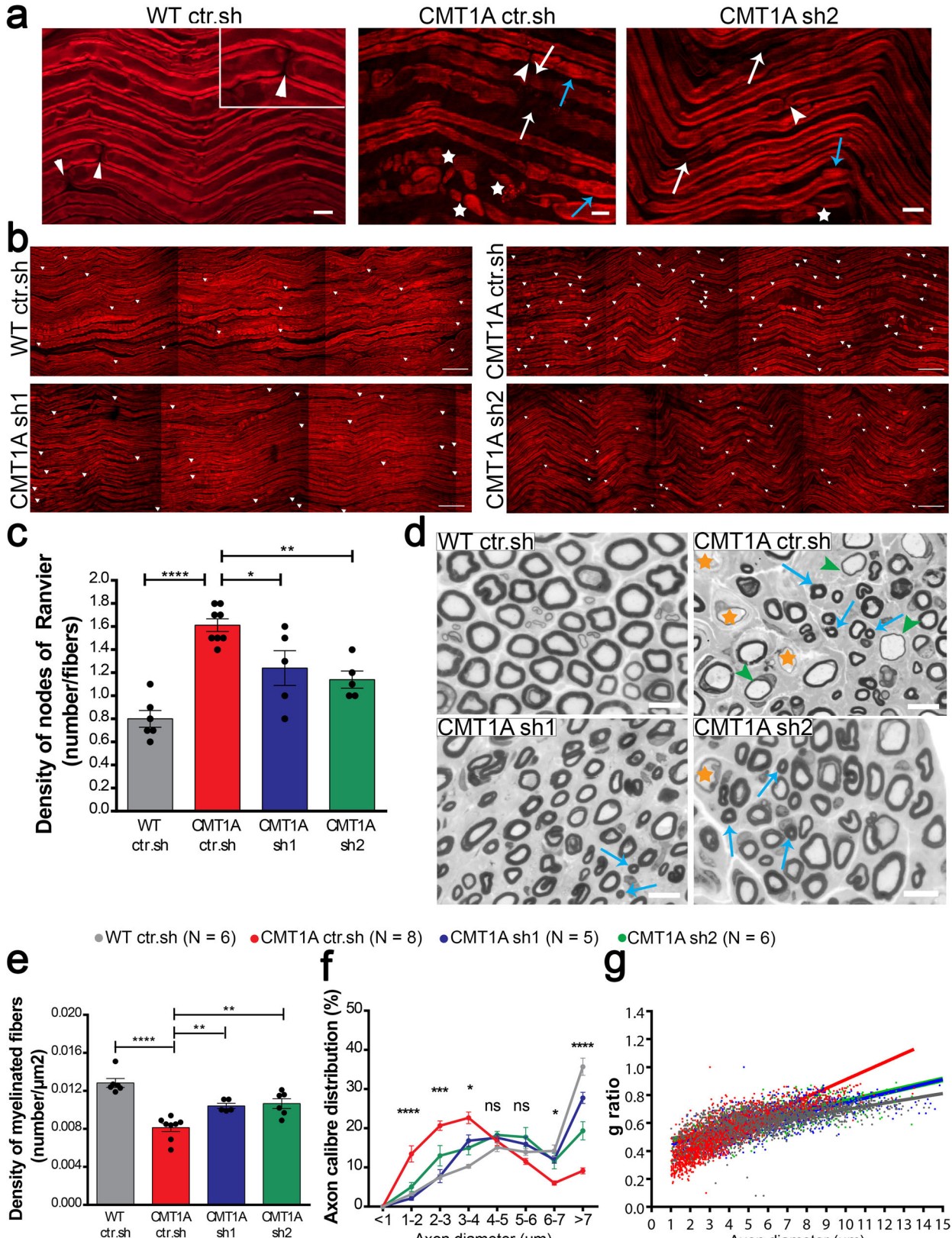

Indeed, when the 3 most relevant biomarkers (*Nrg1.1*, *Gria1* and *Cda*) were selected for the PCA analysis, the segregation between CMT1A ctr.sh and CMT1A sh1 and CMT1A sh2 animals was clear (Fig. 6c). According to the correlation matrix, the grip strength test (4 correlations) was slightly more relevant than Rotarod (3), Randall-Selitto (2) and NCV (1) tests. However, Rotarod (3) and Randall-Selitto (2) tests were sufficient to robustly cover the 3 most relevant biomarkers (Fig. 6b). Taken together these data indicate that a combinative score grouping functional tests and skin biomarkers analysis constitute a reliable and robust measure of CMT1A gene therapy outcome in CMT1A rats.

**Fig. 4 Intra-nerve injections of AAV2/9-sh1 and sh2 prevent the myelin sheath defects in CMT1A rats. a, b** Representative images of CARS imaging on WT ctr.sh, CMT1A ctr.sh, CMT1A sh1 and CMT1A sh2 rat sciatic nerves twelve months post-injection ($n = 6$ animals for WT ctr.sh, $n = 8$ animals for CMT1A ctr.sh, $n = 5$ animals for CMT1A sh1 and $n = 5$ animals for CMT1A sh2). **a** CMT1A ctr.sh nerve shows typical myelin sheath defects such as thin myelin sheath or demyelination (white arrows), focal hypermyelination (blue arrows) and myelin degeneration with myelin ovoids (stars). These defects are less abundant in CMT1A sh2 rat sciatic nerves. The insert represents a zoom of a node of Ranvier (arrowheads). Scale bars: 10 µm and 2 µm for the insert. **b** Nodes of Ranvier were labeled with arrowheads. Nodes of Ranvier are more abundant in CMT1A ctr.sh sciatic nerves compared to WT ctr.sh nerves indicating shorter internodes. CMT1A sh1 and CMT1A sh2 rat sciatic nerves showed less nodes than CMT1A ctr.sh nerves. Scale bars: 50 µm. **c** Graph showing the mean number of nodes of Ranvier on the total number of myelinated fibers per field ($n = 6$ animals for WT ctr.sh, $n = 8$ animals for CMT1A ctr.sh, $n = 5$ animals for CMT1A sh1 and $n = 5$ animals for CMT1A sh2). Statistical test shows one-way ANOVA followed by Tukey's post hoc, two-sided. ****$p < 0.0001$ between WT ctr.sh and CMT1A ctr.sh, *$p = 0.0271$ between CMT1A ctr.sh and CMT1A sh1, **$p = 0.0043$ between CMT1A ctr.sh and CMT1A sh2. **d** Representative images of electron microscopy semi thin sections on WT ctr.sh, CMT1A ctr.sh, CMT1A sh1 and CMT1A sh2 rat sciatic nerves twelve months post-injection. CMT1A ctr.sh nerve shows typical myelinated fiber defects such as large demyelinated axons (orange stars), large hypomyelinated axons (green arrowheads) and small hypermyelinated axons (blue arrows). These defects are less abundant in CMT1A sh1 and sh2 rat sciatic nerve. Scale bars: 10 µm. Graphs showing **e** the mean number of myelinated fibers per area unit (µm²), **f** the mean percentage of axon caliber distribution per axon diameter and **g** g-ratio relative to axon diameter. ($n = 6$ animals for WT ctr.sh, $n = 8$ animals for CMT1A ctr.sh, $n = 5$ animals for CMT1A and $n = 6$ animals for CMT1A sh2). Statistical tests show one-way ANOVA followed by Tukey's post hoc, two-sided (**e**), ****$p < 0.0001$ between WT ctr.sh and CMT1A ctr.sh, **$p = 0.0082$ between CMT1A ctr.sh and CMT1A sh1, **$p = 0.002$ between CMT1A ctr.sh and CMT1A sh2) or two-way ANOVA followed by Dunnett's post hoc, two-sided (**f**), *$p < 0.05$, ***$p < 0.001$, ****$p < 0.0001$ (all $p$-values of two-sided multiple comparison tests are available in the Source Data File); ns, not significant. All error bars represent SEM. Source data are provided as a Source Data file.

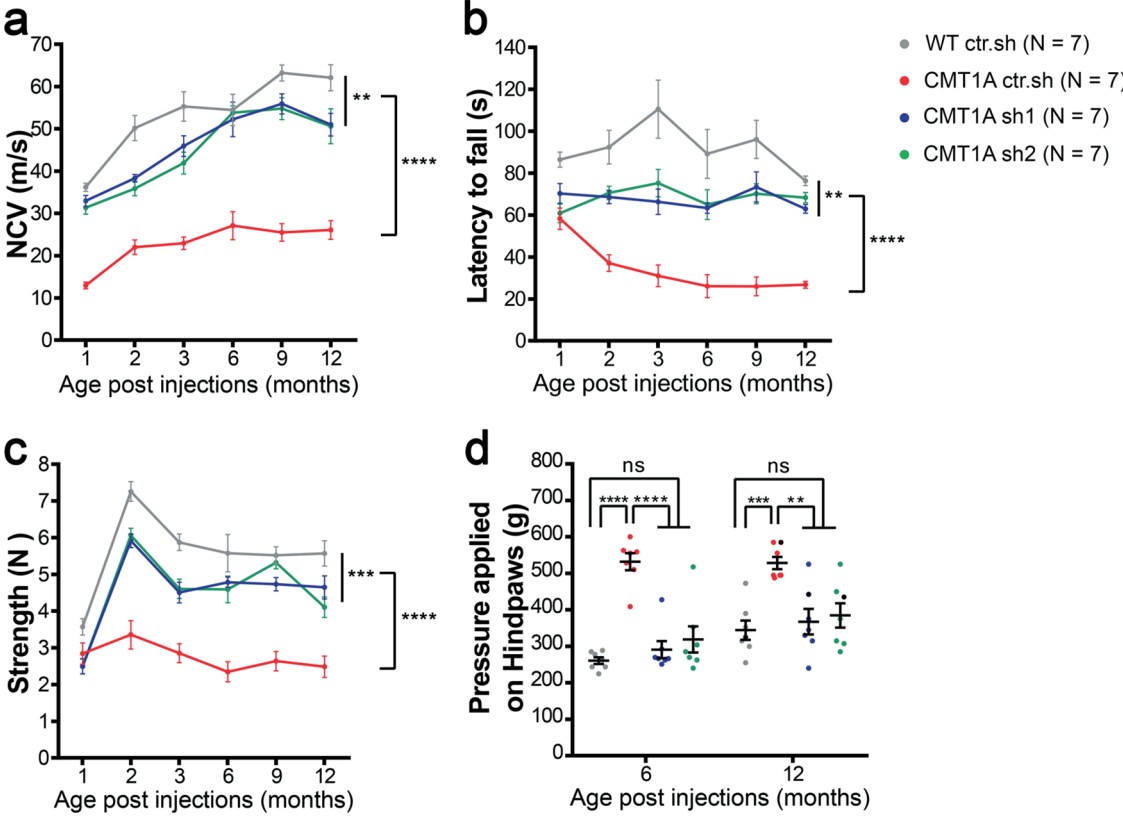

**Fig. 5 Intra-nerve injections of AAV2/9-sh1 and -sh2 prevent motor and sensory defects on the long term in CMT1A rats.** Graphs showing **a** NCV (meter/second), **b** Rotarod test (second), **c** grip test (Newton) one to twelve months after injection and **d** Randall Selitto test (gram) six and twelve months after injection in WT ctr.sh, CMT1A ctr.sh, CMT1A sh1 and CMT1A sh2 animals ($n = 7$ animals per group). Statistical analysis shows two-way ANOVA followed by Tukey's post hoc, two-sided, (**a, b** and **c**) comparing all groups paired two by two (**$p < 0.01$, ***$p < 0.001$, ****$p < 0.0001$, all $p$-values of two-sided multiple comparison tests are available in Table S2) or one-way ANOVA followed by Tukey's post hoc, two-sided, for six and twelve months post-injection in **d**. At six months post-injection, ****$p < 0.0001$ between WT ctr.sh and CMT1A ctr.sh, between CMT1A ctr.sh and CMT1A sh1, between CMT1A ctr.sh and CMT1A sh2; at twelve months post-injection, ***$p = 0.0008$ between WT ctr.sh and CMT1A ctr.sh, **$p = 0.0033$ between CMT1A ctr.sh and CMT1A sh1, **$p = 0.0091$ between CMT1A ctr.sh and CMT1A sh2; ns, not significant. Results are expressed as mean ± SEM. Source data are provided as a Source Data file.

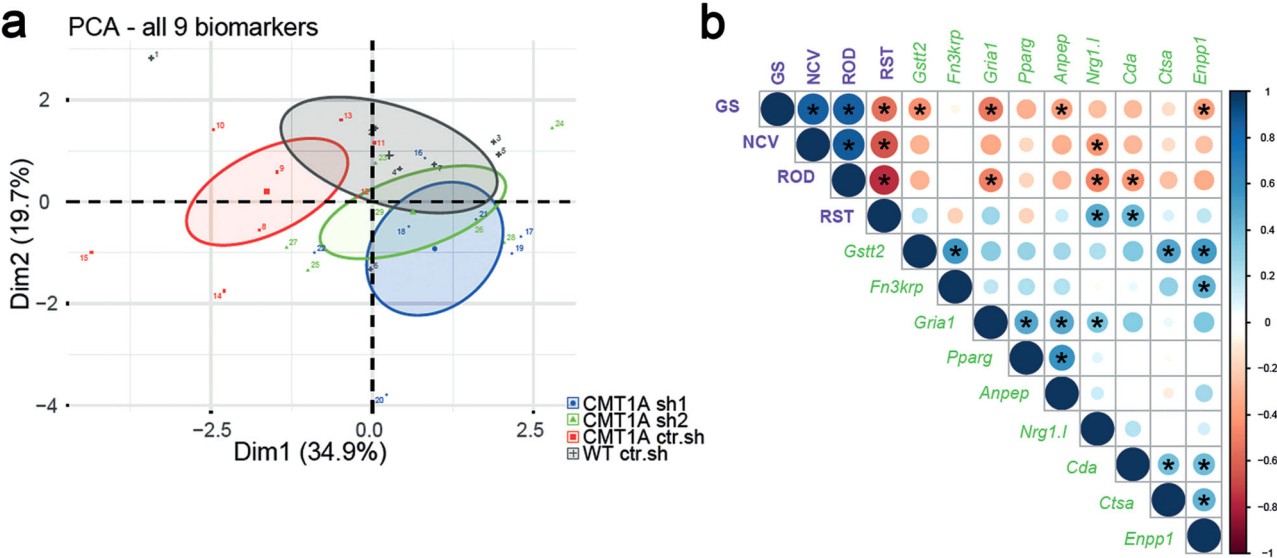

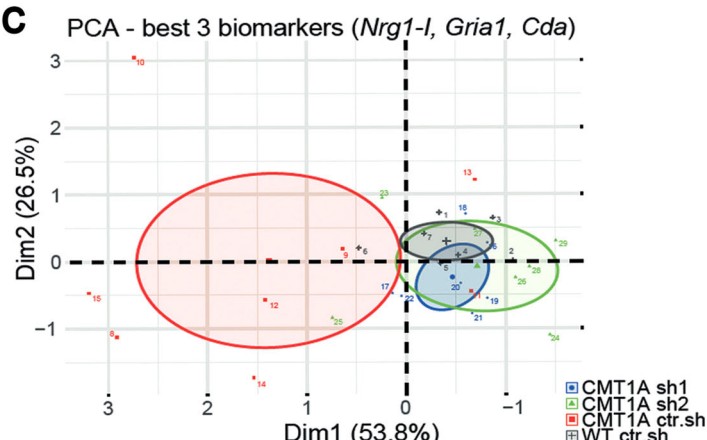

**Fig. 6 Multi-variated analysis of skin biomarkers and sensory-motor phenotypes allows the detection of the therapy outcome. a** Principal component analyses (PCA) of all nine transcriptional biomarkers in forepaw skin biopsies on twelve-month-old animals. Note little to no overlap between WT ctr.sh (gray, $n = 7$) and CMT1A ctr.sh (red, $n = 8$), whereas the treated groups, CMT1A sh1 (blue, $n = 7$) and CMT1A sh2 (green, $n = 7$), show more overlap with the WT ctr.sh group than with the CMT1A ctr.sh group. The mean of each group is given as a center point including the confidence interval (95%) given as an ellipse. **b** Correlation matrix from all animals (total $n = 28$ with $n = 7$ per group) including the expression levels of the skin biomarkers (green labels) and the four functional phenotypic analyses (purple labels): GS, grip strength; NCV, nerve conduction velocity; ROD, Rotarod; RST, Randall-Selitto test. Shown is data from a two-sided Pearson's correlation analyses with graphical representation of the correlation coefficients, from red (−1) to blue (+1) (indicated by circle size and color), and the respective *p*-values (asterisks indicate $p < 0.05$, all the exact *p*-values are available in the Source Data File). **c** Principal component analyses (PCA) of the three best biomarkers (*Nrg1-I*, *Gria1*, *Cda*; see correlation matrix in **b**) in forepaw skin biopsies on twelve-month-old animals (same analysis as in **a**). Source data are provided as a Source Data file.

**Off-target transduction and immune response to the vector are limited after intra-nerve injections**. Regarding a future clinical trial for this gene therapy approach, we investigated the biodistribution of the AAV2/9 vector after its injection in the sciatic nerve of rat pups three months post-injection. Tissues were collected in AAV2/9 injected animals and vector genome copies were analyzed using normalized and standardized qPCR (Table 3). Among the 32 analyzed animals, 29 showed vector genome copies in the sciatic nerves indicating that the injection technique was highly reliable with a success rate of 91%. Among all groups, the average amount of vector genome relative to diploid genome (vg/dg) in the sciatic nerve was $0.54 \pm 0.09$ (Table 3). Four out of 32 animals displayed very low levels of vector genome copies in the liver ($0.005 \pm 0.003$ vg/dg), three in the heart ($0.004 \pm 0.001$ vg/dg) and five in lumbar dorsal root

ganglia L4 L5 ($0.004 \pm 0.001$ vg/dg). One out of 7 animals displayed vector genome copy in the blood (0.005 vg/dg). No vector genome copy was detected in other tested organs (brainstem, spinal cord, kidney, spleen).

This limited distribution of the vector prompted us to measure the immune response against the vector capsid. A validated and standardized ELISA approach was used to measure AAV2/9 neutralizing factors in sera of ten injected animals (5 WT ctr.sh and 5 CMT1A ctr.sh) and four non-injected controls (2 WT and 2 CMT1A rats). Among these samples, only two showed neutralizing factors at low titer (1/500) (Table 4). Taken together these data show that intra-nerve injections of AAV2/9 vector results in a restricted transduction of the injected nerves with limited off-target tissues and a weak humoral immune response toward the vector.

**Table 3 AAV2/9 biodistribution in sciatic nerves and different organs three months after intra-nerve injection.**

| | WT ctr.sh | | CMT1Actr.sh | | CMT1Ash1 | | CMT1Ash2 | | Overall | |
|---|---|---|---|---|---|---|---|---|---|---|
| | Positive/total | vg/dg | Positive/total | vg/dg | Positive/total | vg/dg | Positive/total | vg/dg | Positive/total | vg/dg |
| Sciatic nerve | 8/8 | 0.40 ± 0.31 | 7/8 | 0.21 ± 0.11 | 7/8 | 0.55 ± 0.22 | 7/8 | 1.03 ± 0.86 | 29/32 | 0.54 ± 0.054 |
| Drg L4–L5 | 2/8 | 0.003 ± 0.001 | 2/8 | 0.006 ± 0.004 | 1/8 | 0.003 | 0/8 | NA | 5/32 | 0.004 ± 0.002 |
| Lumbar spinal cord | 0/8 | NA | 0/4 | NA | 0/2 | NA | 0/2 | NA | 0/16 | NA |
| Heart | 2/8 | 0.004 ± 0.001 | 0/8 | NA | 1/8 | 0.003 | 0/8 | NA | 3/32 | 0.004 ± 0.001 |
| Liver | 1/8 | 0.003 | 1/8 | 0.008 | 2/8 | 0.004 ± 0.002 | 0/8 | NA | 4/32 | 0.005 ± 0.003 |
| Spleen | 0/4 | NA | 0/2 | NA | 0/1 | NA | 0/1 | NA | 0/8 | NA |
| Kidney | 0/4 | NA | 0/2 | NA | 0/1 | NA | 0/1 | NA | 0/8 | NA |
| Brainstem | 0/4 | NA | 0/2 | NA | 0/1 | NA | 0/1 | NA | 0/8 | NA |
| Whole blood | 0/3 | NA | 1/2 | 0.005 | 0/1 | NA | 0/1 | NA | 1/7 | 0.005 |

Rats were injected at P6–P7 and sacrificed three months later. We collected 0.5 cm long sciatic nerves located 0.5 cm proximally to the injection site and the other organs. The first column shows the number of positive samples (> LOQ) over all tested samples. The second column shows the transduction rate expressed in vector genome/diploid genome (vg/dg). Results are expressed as the mean ± SD. NA, Not Applicable (< LOQ). Source data are provided as a Source Data file.

**Table 4 AAV2/9 neutralizing factors titration three months after intra-nerve injection.**

| | Positive sera/total | Titer |
|---|---|---|
| WT ctr.sh | 1/5 | 1/500 |
| CMT1Actr.sh | 1/5 | 1/500 |

The first column shows the number of positive sera over all tested samples. The second column shows the titer of positive sera

## Discussion

CMT1A is the main hereditary peripheral neuropathy representing more than 50% of all these significantly disabling diseases[34]. Since 2004, several therapeutic strategies have been proposed and tested preclinically and clinically but as of yet no treatment is available for this disease[4]. At the present time, the most advanced strategy is a pharmacological treatment that reached the clinical phase III (NCT02579759). Another pharmacological treatment is reaching the clinical phase I (NCT03610334) and several others are in preclinical phases[8–10,27,35–37]. As all these pharmacological treatments require a regular and permanent treatment with potential side effects on the long term, gene therapy represents an attractive alternative. Indeed, an indirect gene therapy approach involving the transduction of muscle cells to increase their production of neurotrophin-3 and promote axon survival is in clinical trial phase I/IIa (NCT03520751)[38,39]. Here, we investigated the conditions for a successful gene therapy approach directly targeting mSC, the defective cells in the disease, through an intra-nerve delivery.

Our first goal was to evaluate the transduction efficiency and the specificity of AAV2/9 and 2/rh10 serotypes, which are the main serotypes used to transduce the nervous system, regarding mSC when injected directly in the nerve. In 2015, Tanguy et al.[40] observed a cell transduction in the sciatic nerve of mice intravenously injected with AAV2/9 one day after birth but no detail on the cell type was provided. Hoyng et al. described in 2015 the transduction efficiency of AAV2/1 to 9 in rat and human Schwann cells in vitro and in sectioned mouse nerve segments undergoing demyelination ex vivo[23]. As the existing data were inconclusive, we tested AAV2/9 and AAV2/rh10 serotypes in mouse, rat and NHP in vivo after intra-nerve injections. We found that both serotypes were able to transduce mSC in all these species and at different ages (newborn and adult). AAV2/9 was significantly more efficient than AAV2/rh10 regarding the transduction rate at the injection site (83% vs 32% respectively on average). This high transduction rate is clearly correlated to the injection protocol as an intrathecal injection of the same vectors of newborn or adult mice resulted in the transduction of a large amount of neurons and glial cells[41–44] but no mSC in sciatic nerves. While we cannot rule out a transduction of mSC in nerve roots close to the spinal cord after intrathecal injection, intra-nerve injection appears as the most efficient way to transduce these cells in vivo.

The AAV2/9 specificity for mSC was found to be slightly superior to that of AAV2/rh10 serotype in these conditions (on average 92% vs 88% respectively of transduced cells are mSC). As very few axons were found positive for GFP after AAV2/9 injection in the nerve, further restricting the transgene expression in mSC through a specific promoter appeared redundant.

We found that the intra-nerve injection of $1 \times 10^{11}$ vg AAV2/9 in the rat pup sciatic nerve resulted in an average of 0.54 vg/dg. As AAV2/9 is 95% specific for mSC in our conditions, 87% of mSC are transduced by this vector and knowing that mSC represent 51% of the cells in a sciatic nerve[45], this indicated that

nearly all mSC of the nerve are transduced with one vector copy. This was consistent with the high transduction rate of mSC that we observed using immunolabelling. This also indicated that the vector dose we used to transduce mSC of the rat pups is sufficient. Moreover, the high transduction rate that we obtained in the other species, and in particular in NHP, suggested that we also reached the sufficient dose in these conditions. Further optimizations of the treatment are therefore to be found in the extent of the vector diffusion along the nerve.

We evaluated the diffusion of the vectors when injected in the nerve. Both vectors diffused similarly (at least over 2.5 cm) in the adult mouse sciatic nerve. However, $5 \times 10^{10}$ vg of AAV2/9 distributed in 8 µl was able to transduce on average 83% of mSC along 2.5 cm-length of the entire sciatic nerve, while AAV2/rh10 only transduced on average 45% of mSCs along the same length in the same conditions. This large diffusion of AAV2/9 vector in injected nerves was significantly superior to that of AAV2/8 vector reported by Homs et al. (0.8 cm)[22]. Similar results were obtained for AAV2/9 in the rat pup nerve ($1 \times 10^{11}$ vg in 8 µl transduced on average 73% of mSC over 4 cm) and in NHP nerve ($5 \times 10^{12}$ vg in 416 µl transduced on average 53% of mSC over 6 cm). This represented the total length of rat pups and adult mice sciatic nerves and 30–50% of the adult NHP sciatic nerve. The injection technique that we set up to inject viral vectors in peripheral nerves of adult mice and rat pups[24] is essential in this significant diffusion ability. Indeed, injections are done with a very fine needle under 1.5 to 2 bars pressure by successive small pulses of few nanoliters each. The fine needle allows for perforating the sheath that surrounds the fibers (epineurium and perineurium) while limiting fibers injury. The short and moderated pressure pulses allow injecting a relatively large volume without inducing pressure injury in the nerve. In addition, it prevents the leak of the viral solution out of the nerve due to the immediate tissue resistance to large volume increase. Adapting and using this technique for NHP intra-nerve injection will most likely increase the spread of the vector in this species also. In patients, while transdermal intra-nerve injections are not uncommon in particular during regional anesthesia[46], the direct injection in the nerve is presently undesired because of the toxic nature of concentrated anesthetics and the risk of fiber damages due to the large needles used and the high pressure applied[47]. Therefore, adapting our non-traumatic injection technique for injection of non-toxic vector solution in human nerves via the NHP model would probably be an essential step toward a clinical application of this gene therapy.

The AAV2/9 vector was used to introduce shRNAs into mSC of CMT1A rat sciatic nerves bilaterally in order to decrease PMP22 overexpression. CMT1A rats are transgenic animals that overexpress additional copies of mouse *Pmp22* in mSC in addition to two endogenous copies of rat *Pmp22*[26]. While several mouse models overexpressing human PMP22 are available to mimic the disease in animals[48], the rat CMT1A was chosen here because it mimics more closely the clinical aspect of the disease[28]. Indeed, notwithstanding the discovery of two shRNAs specifically targeting human PMP22 protein expression, our goal was less to characterize a therapeutic product than to evaluate the functional benefit of such a gene therapy directly targeting the molecular and cellular causes of the disease. *In fine* the success of a therapy for CMT1A patients will be less based on the product itself than on the benefit (vs risk) for the patient and on the way we measure this benefit. In this regard, the validation of skin biomarkers as markers of the gene therapy efficiency in CMT1A rats constitutes a progress.

We designed two different shRNAs: sh1 significantly decreases both mouse and rat PMP22 expression while sh2 targets only mouse PMP22 expression. Both shRNAs efficiently and similarly reduced the relative amount of PMP22 protein in CMT1A rat nerves down to that of WT ctr.sh level. As we were unable to distinguish between the mouse and the rat PMP22 proteins, we do not know whether sh1 had a different impact on the mouse or the rat protein expression. However, our data suggest that the reduction of relative PMP22 protein level is not correlated to the species specificity but to the general amount of active shRNA that is expressed in cells. This indicates that the intensity of the downregulation is positively correlated with the amount of vector injected in the tissue. As PMP22 haploinsufficiency and hence under expression is responsible for peripheral neuropathy with liability to pressure palsy (HNPP), the control of the downregulation represents one of the challenges of this gene therapy. Thus, the definition of the maximal safe dose to be injected is another step toward a clinical application.

While we observed a decrease of PMP22 protein expression, we failed to record any decrease in the expression of mouse or rat *Pmp22* mRNA in treated CMT1A rats. The operating mechanism of shRNA that are processed into siRNA by Dicer in cells has been reported to be dual: the interaction of some of these inhibitory molecules with the target mRNA leads to the degradation of the target or to the block of the translation machinery[49]. Our data suggest that both sh1 and sh2 act through the second mechanism.

We treated young CMT1A animals six to seven days postnatal when peripheral nerve myelination is most active because a large amount of nerve defects and of motor impairments result from the alteration of the initial phase of nerve myelination. Indeed, early nerve defects and impairments already occur in young CMT1A rats[27–29]. This is consistent with the disease onset occurring in the first decade in 75% of CMT1A patients[50]. Moreover, the treatment of young CMT1A rats (P6 to P18) with soluble Neuregulin-1 was sufficient to halt disease progression at least until 9 weeks of age, while treatment of adult animals has only a limited impact on the disease[27]. We found that AAV2/9-sh1 and -sh2 treatments significantly increased MPZ protein expression in treated CMT1A nerves suggesting that myelin production is increased. Indeed, morphological analysis indicated that significantly more axons were myelinated and the myelin thickness is slightly increased (g-ratio decrease) in treated CMT1A nerves. Moreover, CARS analysis showed that internodes, the myelinated part of the axon between two nodes of Ranvier, were longer in treated CMT1A animals. As the number of myelinated segments and the length of internodes are determined early during myelination[51], this indicates that our gene therapy prevents the deficit of myelination occurring early on in CMT1A rats. This is confirmed by the NCV analysis: at one month CMT1A ctr.sh rats already have a reduced NCV compared to WT ctr.sh rats due to the deficit of myelinated segment at early stages of postnatal development. The gene therapy is able to prevent this defect as soon as one month, well before impairments appear at the motor behavior level, indicating that the benefit of the therapy occurs through an improved myelination at early stages. While our CARS analysis suggest that AAV2/9-sh1 and -sh2 treatments also prevents late-occurring defects such as focal hypermyelination and segmental demyelination, a benefit of the gene therapy for older diseased animals remains to be shown. Regarding potential future clinical studies, these data suggest that the treatment of CMT1A through gene therapy could entail modulating the expression of PMP22 as early as possible. The possibility to treat CMT1A children in the long term using a gene therapy approach could constitute a major change as all existing pharmacological strategies target adult patients.

Over the past decade AAV-based therapies have shown numerous successes from proof of concept to clinical trials in genetic diseases[11,16]. These studies have also identified off-targets

transduction and humoral immune response against the vector as the two main serious obstacles that hinder successful AAV-based therapies in patients[52–54]. We therefore evaluated the biodistribution of AAV2/9 vector throughout the rat body three months after injection in sciatic nerves. While 91% of the injected nerves showed vector expression, very few animals had this vector in their liver (12%), heart (9%), kidney (0%), spinal cord (0%), spleen (0%) and blood (14%). Moreover, vector genome copies were detected in dorsal root ganglia L4 and L5 for only few animals (16%) despite the fact that these ganglia are located in the close vicinity of the sciatic nerve. As AAV9 are efficiently transported retrogradely when injected in muscles[55], the low amount of dorsal root ganglia and spinal cords positive for GFP mRNA in our experiments suggests that intranerve injections limits the transduction of axons. This biodistribution pattern is unusually limited for an AAV2/9-based gene therapy treatment. Indeed, different delivery routes of AAV2/9 in mammals (mouse, rat, dog and NHP), such as intravascular, intracerebroventricular, intrathecal and intraparenchymal routes resulted in a large amount of transduction in the liver and the heart in the majority of injected animals[44,56–60]. The intra-nerve injection clearly limits the spread of the vector throughout the body probably through the several layers of cells that compose the nerve surrounding sheath[61]. This restricted biodistribution of the vector will potentially prevent off-target side effects that usually plague gene therapy approaches.

Another probable effect of this limited spread of the vector throughout the body is the low immune response that we observed following the treatment. This response, often directed against the AAV capsid, can block AAV transduction if neutralizing factors are pre-existing. Therefore, the immune response toward the injected therapeutic vector is a serious hurdle for gene therapy treatments when considering vector administration or re-administration[52,53]. Fortunately, among all serotypes, AAV serotype 9 exhibits one of the lowest seroprevalence in humans with less than 50% of the population[62]. In this study, 2 out of 10 injected animals presented neutralizing factors against AAV2/9 capsid in their blood. Moreover, the titers of these factors were low suggesting they were not abundant. Whether these factors pre-existed the treatment is not known. In any case, we found no correlation between the presence of these factors and a reduced benefit of the therapy suggesting these low-titer neutralizing factors had no impact on the therapy. In addition, these data suggest that a re-injection of the vector in other nerves of treated animals will not generate an acquired immune response to the vector. This opens the possibility for successive treatments of several nerves in a potential gene therapy strategy for CMT1A.

A clinical trial of this gene therapy approach will require the measure of the outcome of the treatment. Several clinical scores based on functional tests and assessments such as ONLS[63], CMTNS[19,20] or CMTPedS for children[64] exist. However, several clinical trials have shown that these scores remain weakly discriminative regarding a slowly progressive disease such as CMT1A[32,33]. While most of these trials have involved adult patients for whom the disease may progress more slowly than for children, it remains to be seen whether other means are required to measure outcomes. We focused here on skin biomarkers expression described by R. Fledrich and M. Sereda[21,28]. The expression of several genes in skin biopsies of CMT1A rats were used to identify prognostic and disease severity biomarkers, which correlate with clinical impairment. Nine of these genes were then selected as biomarkers of the disease severity in 46 patients[28]. More recently, these biomarkers were further validated in 266 clinically well-characterized genetically proven patients with CMT1A and their use as markers of the disease progression was validated on a 2–3 years interval[21]. We found that, while

individually none of these biomarkers was robust enough, taken together they were able to discriminate between sham-treated and PMP22 shRNA-treated animals. Furthermore, we showed that a multi-variated analysis including functional tests (Rotarod, grip strength, NCV and Randall-Selitto) and biomarkers analysis provide a list of three biomarkers that are sufficient to measure the outcome of the treatment. A combination of two functional tests is also sufficient to cover the full scale of relevant biomarkers. Therefore, we provide here a toolbox to reliably measure the outcome of a treatment in a preclinical study in CMT1A rats. The variable number of significant correlations between biomarkers and functional tests also suggests the scalability of the proposed outcome measure. However, this remains to be confirmed in a protocol using different doses of the treatment. In addition, as these events that are linked and promoted by our treatment, it remains unclear whether the biomarkers changes reflected a higher myelin amount or a higher maintenance of axons. Finally, as we tested animals twelve months after the treatment, it would be useful to know the robustness of the outcome measure in a less favorable situation such as at one or two months post-treatment. In any case, if one considers that our functional tests in CMT1A rats are similar to clinical scores, our data suggest that combining a clinical score with a transcriptional analysis of the three most relevant skin biomarkers will provide a reliable, robust and probably scalable outcome measure of the gene therapy in CMT1A patients.

## Methods

**Study design.** The goal of this study was to first assess the transduction pattern of AAV vectors serotype 2/9 and 2/rh10 in rodents and NHP after intra-nerve injection, and then the efficiency and the safety of a gene therapy approach based on AAV serotype 9 viral vectors expressing shRNA directed against *Pmp22* mRNA in CMT1A rats. The therapeutic readouts analyzed were: downregulation of *Pmp22* mRNA and PMP22 protein levels in sciatic nerves using RT-qPCR and Western blot; upregulation of myelin amount trough the measure of MPZ protein level in sciatic nerves using Western blot; nerve fibers morphological evaluation using CARS, immunohistology and thin section electron microscopy; nerve electro-physiological analysis; motor and sensory behavioral performances; skin mRNA biomarkers analysis using RT-qPCR; multi-variated analysis including skin biomarkers expression and functional phenotypes data; biodistribution of the vectors in several organs using qPCR; vector neutralizing factors tittering in the blood. Experimental groups were sized according to the literature to allow for statistical analysis. No outliers were excluded from the study. Behavioral data originating from animals that died or had physical disabilities unrelated to CMT1A disease during the study were not used. Sample collection, tissue processing and treatment are included and described in the Results and Methods. Rats were randomly assigned to the different experimental groups after genotyping. Scientists who performed the experiments and analysis were blinded to the group's identity. Data were analyzed by those carrying out the experiments and verified by the supervisors.

**Cloning and vector production.** Cloning of the enhanced GFP, mouse and control shRNAs in pAAV and AAV vector productions were provided by the CPV Vector Core of INSERM UMR 1089, Université de Nantes (France). Briefly, ssAAV2/9-CAG-GFP and ssAAV2/rh10-CAG-GFP vectors were obtained from pAAV-CAG-GFP plasmid containing AAV2 inverted terminal sequences, CAG promoter and the enhanced GFP. These vectors were used to determine the transduction pattern following a direct intra-nerve injection. ShRNA sh1 recognizes both Rattus Norvegicus and Mus Musculus *Pmp22* mRNAs while shRNA sh2 only recognizes Mus Musculus *Pmp22* mRNA. Both shRNAs were cloned under the control of U6 promoter in a pAAV plasmid expressing enhanced GFP protein under a CMV promoter (pAAV-sh1 and pAAV-sh2 respectively). These two plasmids were used to generate AAV2/9-sh1 and AAV2/9-sh2 vectors. These two vectors were used to evaluate their efficiency in the CMT1A rat model. A control vector AAV2/9-U6-ctrl.sh-CMV-GFP expressing a shRNA with no target in mammals served as a control (named AAV2/9-ctr.sh).

Vector production was performed following the CPV facility protocol[65]. Briefly, recombinant AAVs were manufactured by co-transfection of HEK293 cells and purified by cesium chloride density gradients followed by extensive dialysis against phosphate-buffered saline (PBS). Vector titers were determined by qPCR, the target amplicons correspond to the inverted terminal repeat (ITR) sequences, ITR-2. ShRNAs targeting human *PMP22* mRNA (shA and shB) were also cloned in pAAV vector as described above (pAAV-shA and pAAV-shB respectively). All primer sequences can be found in Supplementary Table S3.

**Animals included in this study**. All animal experiments were approved by the comité regional d'éthique pour l'expérimentation animale Languedoc-Roussillon and the ministère de la recherche et de l'enseignement supérieur (authorization 2017032115087316 and 2016091313354892 for rodents and 2015061911295753v2 for NHP). All the procedures were performed in accordance with the French regulation for the animal procedure (French decree 2013-118) and with specific European Union guidelines for the protection of animal welfare (Directive 2010/63/EU). Rodents were maintained on a 12 h dark, 12 h light cycle with an ambient temperature of 21–22 °C and humidity between 40 and 60%.

**Mice**. C57BL/6 mice were purchased from Janvier Labs (France). These mice were used to evaluate the transduction pattern of AAV2/9 and AAV2/rh10 after intra-nerve injection. A cohort of three adult mice (2–3 months old) and three pups (P2–P3) were injected with each AAV vectors as described in the section "vector delivery" below.

**Rats**. A breeding colony was established from a gift of CMT1A rats from Max Planck Institute of Experimental Medicine, Goettingen, Germany. Littermates and CMT1A rats were identified using PCR on DNA isolated from the tail[26] (Supplementary Table S3). For the transduction pattern, a cohort of three adult rats (2-3 months old) and three pups (P6–P7) were injected with each AAV vectors as described in the section "vector delivery" below.

For the gene therapy assay, wild-type littermate (WT) and CMT1A male and female rats were randomly divided into four groups (WT ctr.sh, CMT1A ctr.sh, CMT1A sh1, CMT1A sh2) of sixteen rats each. At three months of age half of the rats in each group (eight per group) were sacrificed for biochemical and biodistribution studies. All the others were kept until twelve months of age to study the efficiency of the gene therapy on the sensory-motor behavior and NCV at different time points post injection (1, 2, 3, 6, 9 and 12 months). Finally, these animals were sacrificed for histological and biochemical studies.

**Non-human primates**. Two juvenile cynomolgus macaques (*Macaca fascicularis*, two females, 3.7 years old/ 4.3 kg and 2.3 years old/2.9 kg) were included in the study. Animals were part of the the MIRCen colony (CEA Fontenay-aux-roses, France). Progenitors were imported from a licensed primate breeding centers on Mauritius and Philippines. The experiments were performed in an authorized user facility (Ministère de l'Agriculture, number 92–032-02). Non-human primates remained under veterinary care during the full study. Animals were tested negative for anti-AAV2/9 or anti-AAV2/rh10 antibodies before the treatment.

## Vector delivery

**Mice and rats**. The intra-nerve injection of AAV vectors into the sciatic nerve was performed under anesthesia with isoflurane, the skin on the thigh was disinfected with betadine solution and ethanol 70% and cut above the sciatic nerve location. *Biceps femoris* and *gluteus superficialis* muscles were carefully separated to expose the small cavity containing the sciatic nerve which was lifted with a spatula[24]. Next, the viral solution was injected using a fine glass needle (borosilicate glass capillary GC 100-10, Harvard Apparatus, France) manipulated with a micromanipulator (World Precision Instruments, France). The injection was performed using a pneumatic picopump (PV820, World Precision Instruments, France) controlled by a pulse generator (GW GIG8215A, INSTEK, France). AAV vectors were injected in the sciatic nerve close to the sural nerve branching proximally to the bifurcation between the common peroneal and tibial nerves (0.5 and 1 cm in adult mice and rats and 0.2–0.5 cm in mouse and rat pups, respectively). At the end of injection, the sciatic nerve was placed back inside its cavity, muscles were replaced and the wound was closed with staples (12-020–00, Fine Science Tools, France). Animals were then treated with buprenorphine (100 μg/kg) for two days.

AAV vector solutions were prepared by diluting vectors at the right titer with sterile phosphate-buffered saline and 0.01% Fast Green. For the transduction pattern analysis, pups and adult mice were unilaterally injected with 2 μl containing $1 \times 10^{10}$ vg and with 8 μl containing $5 \times 10^{10}$ vg respectively. Pups and adult rats were injected unilaterally with $1 \times 10^{11}$ vg/nerve in 8 μl and $1.8 \times 10^{11}$ vg/nerve in 30 μl respectively. Control animals were injected with sterile phosphate-buffered saline containing 0.01% Fast Green. For injection in the CMT1A rat model, rat pups were injected bilaterally with $1 \times 10^{11}$ vg/nerve in 8 μl.

**Non-human primates**. Animals were sedated through an intramuscular treatment of ketamine hydrochloride (Imalgene, 10 mg/kg) and xylazine (2% Rompun, 0.5 mg/kg). Anesthesia was then maintained with propofol (Propovet, 10 mg/kg/h, intravenous infusion in the external saphenous vein).

After anesthesia, animals were placed in prone position and a small incision on the skin was performed above the popliteal fossea. Animals were injected in the sciatic nerves 1 cm proximally to the bifurcation between common peroneal and tibial nerves. Both animals received a single and unilateral injection in the left sciatic nerve. A 22 G needle was manually inserted under the epineurium to guide a 150 μm-wide silica capillary 3 mm deep into the nerve. The capillary was connected to a 1 ml Hamilton syringe mounted on an infusion pump (Harvard Apparatus, France) to allow infusion of the vector at 13.9 μl/min. NHP were injected with $5 \times 10^{12}$ vg/nerve in 416 μl. Vectors were diluted in Fast Green (0.005% final

concentration). During the procedure, animals were placed on warming blankets and physiological parameters were monitored. After full recovery from anesthesia, they were replaced in their home cages and clinical observations were daily performed by trained technicians for any sign of discomfort or distress during seven days.

## Tissue collection and processing

**Mice and rats**. Animals were euthanized one month (transduction pattern study) or 3 months (gene therapy assay) post-injection using Pentobarbital (54.7 mg/ml, 100 mg/kg, CEVA Santé Animale, France). They were transcardially perfused with phosphate-buffered saline (PBS) and sciatic nerves were freshly and quickly dissected. Nerves were then fixed for 1 h in 4% paraformaldehyde (PFA)/PBS solution at room temperature or 24 h in 4% PFA/2.5% glutaraldehyde/PHEM (PIPES, HEPES free acid, EGTA, MgCl₂) for electron microscopy analysis. For biochemical studies, tissues were directly frozen in liquid nitrogen and stored at −80 °C.

**Non-human primates**. Euthanasia was achieved through an overdose of Pentobarbital Sodium (180 mg/kg, intravenous injection), four weeks after injection, under sedation by intramuscular injection of ketamine hydrochloride (Imalgene, 10 mg/kg) and xylazine (2% Rompun, 0.5 mg/kg). Animals were then intracardially perfused with 1 l of PBS then 2.5 l of ice cold 4% PFA/PBS. For each NHP, the injected sciatic nerve was examined and collected. Contro-lateral non injected sciatic nerves served as a control. After one night of post-fixation in 4% PFA/PBS, they were processed for histological studies as described in the histological study section.

**Histological study**. Mouse anti-Myelin Basic Protein (MBP-SMI-99, Millipore, reference NE1019, 1/1000), rat anti-Myelin Basic Protein (BIO-RAD, reference MCA409S, 1/1000), rabbit anti- β-Tubulin III (Tuj1, Sigma, reference T2200, 1/1000), rabbit anti-PMP22 (Sigma-Aldrich, SAB4502217, 1/500), rabbit anti-glial fibrillary acidic protein (GFAP, Dako, reference Z0334, 1/1000), rabbit anti-Neurofilament 200 (NF, Sigma, reference N4142, 1/500), mouse anti-E-cadherin (E-cad, BD Biosciences, reference 610182, 1/500), mouse anti-Sodium Channel (pan-Nav, Sigma, reference S8809, 1/500), goat anti-Contactin-1 (CNTN-1, RD System, reference AF904, 1/2000), donkey anti-mouse Alexa Fluor 594 (Thermo-fisher, reference A-21203, 1/1000), donkey anti-rabbit Alexa Fluor 594 (Thermo-fisher, reference A-21207, 1/1000), donkey anti-rat Alexa Fluor 594 (Thermofisher, reference A-21209, 1/1000) donkey anti-rabbit Alexa Fluor 647 (Thermofisher, reference A-31573, 1/1000) and donkey anti-goat Alexa Fluor 647 (Thermofisher, reference A-21447, 1/1000) were used for both immunohistochemistry on frozen sections and on teased fibers.

**Immunohistochemistry on frozen sections**. Following fixation, samples were incubated 24–48 h in two successive baths of 6% and 30% sucrose and then embedded in Optimal Cutting Temperature (OCT, NEG-50, MM France) and stored at −80 °C. Coronal sections (10 μm of thickness) were cut using cryostat apparatus (LEICA CM3050). Cryosections were blocked with 5% Normal Goat Serum (NGS) and 0.1% triton X-100 in PBS, incubated overnight at 4 °C with primary antibodies diluted in NGS/Triton/PBS, washed with PBS and then incubated 1 h at room temperature with secondary antibodies diluted in NGS/Triton/PBS. After several PBS washes, cryosections were mounted in Dako fluorescent mounting medium (S3023). AxioScan slide scanner and an Apotome fluorescence microscope (Zeiss, France) were used to obtain images. The percentage of transduced mSC (GFP and MBP positive cells surrounding Tuj1 positive axons) over all mSC (MBP positive but not GFP positive cells surrounding Tuj1 positive axons) in the full section was calculated using Zen software (Zeiss, France). For each species, this percentage was calculated at the injection site, proximally (toward the spinal cord) and distally (toward the paw) regarding the injection site (see Supplementary Fig. S5). Proximally distances of the injection site were 2 cm for mice, 3 cm for rats and 2 and 4 cm for NHP. Distally distances of the injection site were 0.5 cm for mice, 1 cm for rats and 2 cm for NHP.

**Immunohistochemistry on teased fibers**. After sciatic nerve fixation, fibers were gently teased, dried on glass slides, and stored at −20 °C. Teased fibers were permeabilized by immersion in –20 °C acetone for 10 min, blocked at room temperature for 1 h with NGS/Triton/PBS, then incubated overnight at 4 °C with primary antibodies diluted in NGS/Triton/PBS. Slides were then washed several times with PBS and incubated with secondary antibodies diluted in NGS/Triton/PBS 1 h at room temperature. Slides were mounted on coverslips with Dako fluorescent mounting medium (S3023) and examined using an Apotome fluorescence and a Confocal LSM 880 Airyscan microscopes (Zeiss, France). For each sciatic nerve, 9–10 fields were captured and the percentage of mSC (GFP and MBP positive cells), nmc (GFP and GFAP positive cells) and axons (GFP and NF positive cells) transduced by AAV vectors among all transduced cells were calculated using Zen software (Zeiss, France).

**Electron microscopy of sciatic nerve**. Fixed nerves were then processed by the Electron Microscopy platform at the Institute of Neurosciences in Montpellier

(INM, MRI-COMET). Samples were post-fixed with osmium 0.5% and K4FeCN6 0.8% 24 h at room temperature in the dark, went through dehydration in successive ethanol baths and embedding in Epoxy resin (EMbed-812 Embedding Kit, Electron Microscopy Sciences, France) using a Leica AMW machine. They were finally incubated 36 h in an oven at 60 °C. Nerves were cut with Leica Reichert Ultracut S ultramicrotome into semi-thin sections of 700 nm—1 μm. Sections were stained with Toluidine Blue and imaged using an AxioScan slide scanner (Zeiss, France). The total number of myelinated axons, the mean axon diameter (100 axons per nerve randomly distributed throughout the entire section) and the g-ratio (axon diameter over the full fiber diameter) were measured using ImageJ software and GRatio plugin.

**Cell culture and transfection.** The efficiency of sh1 and sh2 was examined in vitro using Schwann cell lines from mouse (MSC80)[66] and rat (RT4-D6P2T). 500 000 cells per well were seeded in a 6-well-plate in Dulbecco's Modified Eagle Medium (DMEM) (Gibco/Thermo Fisher, France) with 10% Fetal Bovine Serum (FBS) (Gibco/Thermo Fisher, France) and 1% Penicillin Streptomycin (Gibco/Thermo Fisher, France). Twenty-four hours after seeding, cells were transfected using jet-PRIME reagent (Polyplus-transfection S.A, France) according to manufacturer protocol with pAAV-ctr.sh or sh1 or sh2. Two days after transfection, each well was washed three times by PBS and proteins were extracted using RIPA lysis buffer completed by protease inhibitors (Fisher Scientific, France) for 1 h at 4 °C. Three independent experiments were performed with each cell line. The efficiency of shA, shB, sh1 and sh2 to silence human PMP22 expression was done by transfecting HEK293 cells with pCMV3-hPMP22-Flag (HG14519-CF, Sinobiological) and pAAV vectors expressing shRNAs as described previously. Experiments were done as described above.

**Western blot.** Frozen nerves were crushed with a pestle and mortar cooled on dry ice, solubilized in RIPA lysis buffer completed with protease inhibitors (Fisher Scientific, France) and homogenized on a rotating wheel at 4 °C for 3 h. They were then sonicated three times during 10 s on ice (Microson ultrasonic cell disruptorXL, Microsonic) and centrifuged for 30 min at 5300 $g$ at 4 °C. Proteins concentrations from cell lysates or nerve lysates were quantified using the Bicinchoninic acid (BCA) protein assay kit (Thermo Scientific, France). Ten micrograms of proteins was loaded on a 4–20% precast polyacrylamide gels (Mini Protean gels, Bio Rad, France). Proteins were transferred to nitrocellulose membranes (Bio Rad Trans blot transfer pack) through semi-dry transfer process (Bio Rad Bio Rad Trans-Blot Turbo system). Membranes were first incubated with the REVERT total protein stain solution (LI-COR Biosciences, France) to record the overall amount of protein per lane[67]. Then, membranes were blocked for 1 h at room temperature using LI-COR blocking buffer (Odyssey Blocking buffer, LI-COR Biosciences, France). They were incubated with the following primary antibodies overnight at 4 °C in the same blocking buffer: rabbit anti-PMP22 (Sigma-Aldrich, SAB4502217, 1/750), mouse anti-β-Actin Clone AC-15 (Sigma-Aldrich, A1978, 1/10 000), goat anti MPZ (Thermo Fisher Scientific, PA5-18773, 1/1000), rabbit or mouse M2 anti Flag (Sigma-Aldrich, F7425, F1804, 1/1000). Following three washes of 10 min with TBS-0.1% Tween (TBST), secondary antibodies were used at a 1/15 000 dilution in LI-COR blocking buffer: IRDye 800CW donkey anti-rabbit (LI-COR Biosciences, 925-32213), IRDye 680RD donkey anti-mouse (LI-COR Biosciences, 925-68072) or IRDye 800CW donkey anti-goat (LI-COR Biosciences, 925-32214). After three washes in TBST, results and quantifications were obtained by the Odyssey CLX LI-COR Imaging System and its "Image Studio" software.

**Pmp22 mRNA expression in injected sciatic nerves.** Total RNA was isolated from sciatic nerves using TRIzol® reagent (Thermo Fisher Scientific) according to the manufacturer's instructions. Total RNA (150 ng) was treated with RNAse-free DNAse (ezDNAse, Thermo Fischer Scientific) and then reverse transcribed using SuperScript IV VILO Master Mix (Thermo Fischer Scientific) in a final volume of 20 μl. qPCR were conducted on a StepOne Plus™ Real Time PCR System (Applied Biosystems®, Thermo Fisher Scientific) using 50 ng of cDNA (diluted 1/100) in duplicate. Amplifications were performed (30 s at 95 °C, 40 cycles of 5 s at 95 °C, 30 s at 60 °C (*Mpz*, *Actb* and *Rps9* primers) or 61 °C (Total *Pmp22* primers) or 64 °C (Mouse and rat *Pmp22* primers) using SYBR qPCR Premix Ex Taq (Takara). Each sample (5 μl cDNA) was measured in duplicates and averaged. The efficiency, linearity and absence of qPCR inhibition were determined by analyzing serial dilutions of cDNA sample (1/10 to 1/100,000) obtained from a sciatic nerve of the CMT1A ctr.sh group. Quantification of PCR product was performed using the comparative ΔCt method (Relative quantification, RQ). The results were expressed as *Pmp22 / Mpz* relative mRNA expression in the sciatic nerves for each group. The limit of quantification (LOQ) of our test was for mouse-specific *Pmp22* RQ = 0.002, for rat-specific *Pmp22* RQ = 0.001 and for total *Pmp22* RQ = 0.001. *ActB* and *Rsp9* were used as housekeeping genes. All primer sequences can be found in Supplementary Table S3.

**CARS imaging.** All CARS images were obtained with a two-photon microscope LSM 7 MP coupled to an OPO (Zeiss, France)[25] and complemented by a delay line[68]. A ×20 water immersion lens (W Plan Apochromat DIC VIS-IR) was used

for image acquisition. Four consecutive fields (250 mm square, 200 μm deep) were captured on sciatic nerves and an image of 1 cm long/20 μm deep was then reconstructed using Zen software (Zeiss, France). The number of nodes of Ranvier was determined for each reconstructed image. The results were expressed as a density of nodes of Ranvier over the number of fibers per field.

**Behavioral analysis**

*Rotarod.* The rotarod test was performed using the "Rota Rod" machine specific for rats (Bioseb, France). Following one day of training, the latency to fall from the rotating bar was recorded the next day with an acceleration from 4 to 40 rpm over a period of 5 min. Each animal underwent three trials. Data were averaged for each rat and for each group.

*Griptest.* The grip test was performed on the rear paws using a grid connected to an electronic device recording the force in Newtons (Bioseb, France). The muscular strength was measured by recording the maximum amount of force maintained by the animal after it gripped the grid with its rear paws and while it was pulled down by its tail. Each animal underwent three trials. Data were averaged for each rat and for each group.

*Randall Selitto paw-pressure vocalization test.* Mechanical pain sensitivity was measured as the threshold to a noxious mechanical stimulus[69]. Briefly, for two weeks before the experiments, animals were daily placed in the experimental room and were left for 1 h to become accustomed to the environment. Then, they were gently handled for 5 min and exposed to the nociceptive apparatus without stimulation. To evaluate the nociceptive threshold, a constantly increasing pressure was applied to the rat hind paw until vocalization occurs. A Basile analgesia meter (stylus tip diameter, 1 mm; Bioseb, France) was used. A 600-g cut-off value was determined to prevent tissue damage. Results are expressed as the mean of each hindpaw per rat.

**Electrophysiology analysis.** Nerve conduction velocity (NCV) was measured on both sciatic nerves of anaesthetized rats (isoflurane, AST-00 manually operated workstation, Anesteo, France) placed on a heating plate at 37 °C. Proximal and distal stimulations were performed using a pair of 12 mm-steel needle electrodes with 2 mm pin plugs (AD Instruments, MLA1304, Oxford, UK) and were recorded from the intrinsic foot muscles using 12 mm-steel electrodes with 1.5 mm safety socket plugs (AD Instruments, MLA1303, Oxford, UK) placed on the rat's rear paw's plantar muscle and on the middle toe's muscle. A biphasic stimulation lasting 0.2 ms was applied using a PowerLab 26 T generator (AD Instruments, Oxford, UK) connected to LabChart software (AD Instruments, Oxford, UK). Stimulations were delivered with an increasing current intensity until supramaximal stimulation. NCV was calculated as the ratio of the distance between the proximal and distal sites of stimulation in meter on the action potential latency in second using LabChart software.

**Biomarker studies.** Glaber skin of front paws were collected on animals sacrificed twelve months post-injection. Samples were snap-frozen in liquid nitrogen and stored at −80 °C for biomarker analysis[21]. Briefly, total RNA was extracted using RNeasy Mini Kit (Qiagen, France) according to manufacturer's recommendations for non-fatty tissue and precipitated in ice-cold ethanol. The Agilent integrity check was used to verify RNA quality. Samples with an integrity number higher than seven were used for cDNA synthesis using the Superscript III RT kit (Invitrogen, Germany). Real-time semiquantitative PCR with TaqMan and SYBRGreen were performed in the LightCycler 480 Systems (384-well format, Roche Applied Science, Germany) with a reaction mix prepared to the final volume of 10 μl. All reactions were run in four replicates. The threshold cycles for each gene of interest were normalized against the stable housekeeping gene *ActB* (Supplementary Table S3). The PCA and correlation matrix were performed using R-package ade4. For data plotting and statistical analysis RStudio Version 1.0.153 was used. Principal component analysis (PCA) was done with the prcomp function of the stats package (R Core Team (2013) R: A Language and Environment for Statistical Computing. R Foundation for Statistical Computing, Vienna. http://www.R-project.org/). For visualization and plotting of PCA data the factoextra package was used (Alboukadel Kassambara and Fabian Mundt (2017). factoextra: Extract and Visualize the Results of Multivariate Data Analyses. R package version 1.0.5. https://CRAN.R-project.org/package=factoextra).

**AAV2/9 biodistribution and vector copy number.** Samples were collected from rats three months after injection. We collected sciatic nerves 0.5 cm proximally to the injection site over a distance of 0.5 cm, the lumbar dorsal root ganglia 4 and 5 (DRG L4 and L5), the lumbar spinal cord, the heart, the liver, the spleen, the kidney, the brainstem and the blood. Whole blood was collected in tubes containing EDTA. All samples were collected in DNA-free, RNAse/DNAse-free and PCR inhibitor-free certified microtubes. Tissue samples were obtained immediately after sacrifice in conditions that minimize cross-contamination and avoid qPCR inhibition[70]. Samples were snap-frozen in liquid nitrogen and stored at −80 °C. Extraction of genomic DNA (gDNA) from 200 μl of whole blood or from tissues using the Gentra Puregene kit and Tissue Lyser II (Qiagen, France) was performed

accordingly to manufacturer recommendations. qPCR analyses were conducted on a StepOne Plus apparatus (Applied Biosystems®, Thermo Fisher Scientific, France) using 50 ng of gDNA in duplicates. All reactions were performed in a final volume of 20 μl containing template DNA, Premix Ex Taq (Ozyme, France), 0.4 μl of ROX reference Dye (Ozyme, France), 0.2 μmol/L of each primer and 0.1 μmol/L of Taqman® probe (Ozyme, France). Vector genome number was determined using a primer/FAM-TAMRA probe combination designed to amplify a specific region of the *GFP* transgene. Endogenous gDNA copy numbers were determined using the following primers/JOE-TAMRA probe combination, designed to amplify the rat *Hprt1* gene (Supplementary Table S3). For each sample, threshold cycle (Ct) values were compared with those obtained with different dilutions of linearized standard plasmids (containing either the *GFP* expression cassette or the rat *Hprt1* gene). The absence of qPCR inhibition in the presence of gDNA was checked by analyzing 50 ng of gDNA extracted from tissues samples or blood from a control animal. Results were expressed in vector genome number per diploid genome (vg/dg). The limit of quantification (LOQ) was determined as 0.002 vg/dg.

**AAV2/9 neutralizing factors**. The detection of AAV2/9 neutralizing factors was performed by the Gene Therapy Immunology Core (GTI) at INSERM UMR 1089 laboratory (Nantes, France). Briefly, the neutralization test consisted in an in vitro cell transduction inhibition assay using a standard recombinant AAV2/9 expressing the reporter gene system LacZ. Serum samples were collected from rats 3 months post injections. Each serum was tested using a range of dilutions 1/50, 1/500, 1/5000, 1/50,000 and 1/500,000. Gene expression was measured in cell lysates using a chemoluminescent substrate of galactosidase (GalactoStar kit, Invitrogen). Neutralizing factor titer corresponds to the last dilution leading to the inhibition of more than 50% of the standard AAV2/9 transduction (100% corresponds to the maximum transduction obtained with the standard AAV2/9 alone). Five rats injected with AAV2/9-ctr.sh and fives CMT1A rats injected with AAV2/9-ctr.sh were randomly analyzed for the detection of AAV2/9 neutralizing factors.

**Statistical analysis**. Data were analyzed with Graphpad Prism version 7 (Graphpad Software) and expressed as the mean ± standard error of the mean (SEM) or ± standard deviation (SD) as indicated in the Figures legends. Statistical differences between mean values were tested using one-way or two-way ANOVA analysis followed by Tukey's, Dunnett's or Sidak's multiple comparison test, two sided, as indicated in the Figures legends. Differences between values were considered significant with: $*p < 0.05$, $**p < 0.01$, $***p < 0.001$, $****p < 0.0001$. All exact $p$-values are available in the Source Data File.

**Reporting summary**. Further information on research design is available in the Nature Research Reporting Summary linked to this article.

## Data availability
All data supporting the findings of this study are provided within the paper and its supplementary information. Materials are available upon request. The source data underlying Tables 1, 2 and 3, Figs. 2, 3, 4c, e-g, 5a–d and 6, and Supplementary Figs. S6a-c, S7, S9 and Supplementary Table S1 are provided as a Source Data file. All exact $p$-values are available in the Source Data File. Source data are provided with this paper.

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

## Acknowledgements

We thank the SMARTY platform (RAM - Animal facility network of Montpellier) for expert care of animals and help for behavioral tests (Anne-Laure Bonnefont), the Electron Microscopy platform (Chantal Cazevieille), Montpellier Ressources Imaging (Hassan Boukhaddaoui), CEA MIRCen Institut (CEA Fontenay aux Roses, France), CPV vector core and Gene Therapy Immunology core from the INSERM UMR 1089, University of Nantes (https://umr1089.univ-nantes.fr/facilities-cores/). We also are grateful to Klaus-Armin Nave and Mickael Sereda for their gift of CMT1A rat. Funding: This work has been supported by European Research Council grant (FP7-IDEAS-ERC project 311610) and E-Rare program (project CMT-NRG, 11-040) to NT, Labex EPIGENMED through a fellowship to SG and by Droguerie Mercury S.A.L through a fellowship to HH.

## Author contributions

Conceptualization: B.G. and N.T.; Formal analysis: B.G., H.H. and R.F.; Funding acquisition: P.A. and N.T.; Investigation: B.G., H.H., S.S., J.B., M.D., S.A., G.C., C.M.F., A.J., C.R., M.Z., V.F.L.R., C.L.G., V.S., R.F., M.C.C. and S.G.; Methodology: H.H. Project administration: B.G. and N.T.; Resources: B.G., C.M.F., V.F.L.R., C.L.G., A.J., C.R., P.A. and R.F.; Supervision: N.T.; Validation: B.G., H.H., R.F. and N.T.; Visualization: B.G., H.H., R.F. and N.T.; Writing – original draft: B.G., H.H. and N.T.; Writing – review & editing: all co-authors.

## Competing interests

B.G., N.T. and P.A. are inventors of patent WO2017005806A1. The remaining authors declare no competing interests.
