## [Peer Review File · Nature Communications]

Reviewers' Comments:

Reviewer #1:

Remarks to the Author:

In this paper Gautier et al. evaluate the specificity and efficiency of intra-nerve injection of (AAV)-based to silence PMP22 expression using in a transgenic rat model of Charcot-Marie-Tooth disease. The authors evaluated the therapeutic efficiency of the AAV2/9 shRNAs targeting PMP22 mRNA on the human skin molecular biomarkers used to predict CMT1A disease outcome. This study shows impressive correlation results between functional measures and molecular biomarkers using a multi-variated analysis. This study shows high transduction rate of the AAV2/9 shRNAs, restricted to the injected nerves with long term benefits on rodent or NHP peripheral nervous system. This study shows very encouraging results on rodents and NHP that could be translated to human gene therapy research aiming to develop therapies for patients suffering from CMT1A, with potential application to other peripheral neuropathies.

This study is very convincing.

I have only minor points and clarifications that the authors need to address in some of their figure conclusions.

- The supporting figures and tables for the first paragraph on broad and specific transduction are missing an accurate method. It is unclear how the total mSC and transduced mSC were counted. Fig1 a. and b., show only eGFP on cross sections, where is the mSC counterstain? Was it using CARS and Tuj1?

Please clarify this in text/legend and Methods.

- Fig.1 legend. And Table1 legend.

Proximal distances from the injection site were 2, 3 and 4 cm for mice, rats and NHP respectively. Distal distances from the injection site were 0.5, 1 and 2 cm for mice, rats and NHP respectively. I have hard time to understand the nomenclature used as 'proximal' the farthest from the site of injection and 'distal' the shortest from SI?

Since the site of injection have always the highest transduction rate, the proximal transduction rate that is near the SI should be close the SI rate? and the distal rates should be higher?

There is a misunderstanding on what is proximal and distal to SI, please clarify/correct.

- In the methods, there is no information on the injection site in the rodents while it is detailed for the NHP.

- "The transduction rate was between 21% and 69% at these distant points (Table 1). The difference resulted

from the direction of the needle when inserted into the nerve."

By just reading this sentence, without looking at the table, it seems that you have a big variability on distal transduction within animal (from 21% to 69%) due to the inconsistent direction the needle was inserted.

I would rephrase this sentence making a general observation for adult rodents (63% and 91%) and NHP (21% to 69%). The highest transductions rates are observed at the injection site and proximal to the SI and are getting lower distally from the SI.

- Figure2a. & c.

In addition, typical morphological characteristics of mSC were seen through eGFP labeling such as Schmidt-Lanterman incisures (Fig. 2a, arrows), Cajal's bands (Fig. 2a, arrowheads) and nodes of Ranvier (Fig. 2a, stars).

Right panels: higher magnification showing the immunostaining for myelin MBP (red) of teased fibers. eGFP (green) is expressed in the nucleus (star), the Cajal's bands (arrowheads) and the Schmidt-Lanterman incisures (arrows) of mSC.

Some conclusions on subcellular localization of eGFP made from Fig2.a and c. are overstated

regarding eGFP labelling in SLI, nodes of Ranvier, and nuclei.

The authors should revised their conclusions based on the current figures or add additional evidences using appropriate co-staining for SLI, nodes of Ranvier and nuclei.

- Figure4.a

Why there is no loading control to correct quantification loading variability?

Is the variability seen in CMT1A ctrl.sh related to the variability in PMP22 overexpression in CMT1A rat model or a technical variability?

- Figure6.b

With the actual image magnification, it is hard to identify the nodes of Ranvier indicated by white arrow heads. A magnified insert is necessary to show how the node of Ranvier were identified using CARS.

- Figure7a.

When CMT1A animals were treated with AAV2/9-sh1 or sh2, the NCV remained not significantly different to WT ctr.sh values at all-time points for at least 12 months (Fig. 7a).

That is not what the statistics on the graph are showing (**). For the NCV at least 2, 3, 9 and 12mths seem different from WT ctr.sh animals.

Please correct accordingly in the text.

Dr.Belin Sophie

Reviewer #2:

Remarks to the Author:

Charcot-Marie-Tooth diseases are the most common inherited disorders affecting peripheral nervous system (prevalence of 1 in 2500). Among them, CMT1A, an autosomal dominant demyelinating form of CMT caused by a uniform 1.5Mb tandem duplication on chromosome 17p accounts for 70% of demyelinating CMT cases and one third of all CMT cases and affects more than two million people worldwide. Since we have no cure for CMT1A disease, the goal to define new therapies is of an extreme importance.

Gautier and colleagues in their manuscript described a novel approach based on gene therapy as treatment for CMT1A. The animal model used for their study is the well-established line of rat for CMT1A, but also mice and non-human primates were used to test the viral-based approach.

In general, the rationale of this study, the description of the methods and the objective of this research are well defined. The manuscript is also well written.

After reviewing the manuscript of Gautier and colleagues I have few points that in my opinion the authors should address:

1 - The biggest discrepancy that I noticed in this work is different efficiency of the construct "sh2" in silencing Pmp22. It seems that sh2 is able to downregulate Pmp22 in vivo (rat model) but not in vitro in rat cell line (figure 3c and also strongly stated in "Material & Methods" pag. 16 "Cloning and vector production"). Perhaps the cellular model RT4-D6P2T was not the most appropriate and primary culture of rat Schwann cells could be used for a further analysis.

2 - I think that the choice of MPZ as normalizer for the experiment shown in figure 4 (a and b) is not the most appropriate. The authors should have used a glial marker (Sox10? S100b?) or an ubiquitous expressed marker (Beta-actin? Alpha-tubulin?). MPZ expression in the different animal groups is indeed an interesting data and should be maintained in this work. Also in Figure 5B I would recommend to use one of the aforementioned marker instead/in parallel to the total protein quantification.

3 - Some of the bars used for the statistics are not clear. In detail:

- Fig 4b (CMT1A Vs sh2 is highly significant. I assume that also CMT1A Vs sh1 is the same but it is not clear from the graph.

- Fig. 7b and 7C (is wt Vs sh1 AND sh2 highly significant? Or only Wt Vs one the the condition?)

4 – In Figure 6D a representative picture for CMT1A sh2 is needed. And if the space for the figure is sufficient also an example of CMT1A sh1 for Fig. 6A and 6B (this last 2 pictures are to be considered optional).

5 – Some of the biomarker with a different expression showed in Fig.8 (e.g. Nrg1-1) might reflect a preservation of axons after the viral transduction. Authors should comment this or provide evidences of lack of axonal loss in CMT1A rat model after the silencing of Pmp22.

6 – Authors should defend their decision to have chosen the rat CMT1A model instead of other existing murine models, in particular relating the Pmp22 mRNA overexpression observed in human patients and the different animal models. Being Dr. Fledrich R one of the co-author it will not be difficult to address this point (I cite here 2 reviews that could be included in the discussion: "Murine therapeutic models for Charcot-Marie-Tooth (CMT) disease" BMB 2012 and "A rat model of Charcot-Marie-Tooth disease 1A recapitulates disease variability and supplies biomarkers of axonal loss in patients" Brain, 2012).

7 – Figures 1, 3 and 4 all the Tables might be considered as supplementary material if the format for the journal allows it.

Reviewer #3:

Remarks to the Author:

CMT1A is the most common type of inherited demyelinating neuropathy resulting from duplication and overexpression of the PMP22 gene. This paper by Gautier et al., describes the development of a gene therapy approach to treat CMT1A by silencing overexpressed mouse PMP22 in a rat model of the disease using AAV mediated delivery of shRNA via direct injection into the sciatic nerve. This is an important study as it supports the potential use of gene silencing therapy to treat CMT1A. Extensive work has been done and the results are encouraging. However, there are some major and several other issues with the study:

Major concerns:

1. There is limited potential for clinical translation of this approach with direct intraneural injections due to the invasiveness of the procedure and high demand on neurosurgical expertise.
2. The proof-of concept is provided on silencing the mouse PMP22 by targeting the coding region. How can this be translated to treating patients with overexpression of human PMP22? Further experiments to validate the same approach for human PMP22 silencing is needed.
3. Gene therapy in the CMT1A model was tested very early (P6-7), before the onset of the neuropathy. There is no evidence that this therapy would be beneficial after the onset of the neuropathy, which is a major issue in clinical translation. A proof of efficacy after onset in the disease models is needed.
4. Related to points 1 and 3, how do the authors evaluate the potential for regulatory approvals to test such invasive treatment in younger children with none or minimal manifestations?

Further concerns:

1. In the abstract the statement "...widespread transgene expression in myelinating Schwann cells in mouse, rat and nonhuman primate" is misleading and should be re-stated, as we only see expression in short segments of sciatic nerves and basically in no other PNS tissues. In the last paragraph of the introduction they admit that "the dispersion of the vector remained limited to the injected nerves"
2. In the introduction second paragraph, the prevalence for CMT1A of 5-10/10000 is too high, did the authors mean 5-10/100,000? (which would agree with the reference cited)
3. At the end of page 6 it is stated that: "No downregulation beyond that of control levels was observed in CMT1A sh1 and CMT1A sh2 animals" How do the authors explain the lack of silencing also the endogenous rat Pmp22, since sh1 was effective also on the rat gene in vitro?"

4. Related to this, a more detailed and specific investigation of mouse and rat PMP22 silencing effects should be done (for example using real time PCR with specific probes) to understand the degree of lowering expression of endogenous compared to overexpressed gene. This is also essential before discussing any HNPP-like changes resulting from excessive silencing, that can be seen only if endogenous rat PMP22 is also silenced.
5. The method of determining expression in specific cell types is not clear. What markers of non-myelinating Schwann cells were used? Did the authors look for other cell types (fibroblasts, epineurial cells?)
6. The authors should include negative control images for all their expression analysis (Figures 1 and 2). Negative control should be tissue from animals injected with only fast green dye solution (which was mixed with the vector)
7. Please include the missing data of proximal and distal expression in rat nerves (Table 1). Where they low or not done? This is not mentioned in the text.
8. Please include missing data for NHP cell expression specificity (Tables 1 and 2)
9. Please explain in the methods where exactly the quantification of expression rates was done in relation to the injection site in each species
10. Please clarify in methods and results from where in relation to the injection side where the samples taken for VCNs determination.
11. Please clarify in the methods (Vector delivery section) whether unilateral or bilateral injection into the sciatic nerve was performed for every experiment. This is confusing.
12. How can the result of lower tropism of AAVrh10 compared to AAV9 for NHP Schwann cells (Figure 1a) be based on a single injection of a single animal? Do the authors feel that this is a conclusive finding? Please discuss this limitation.
13. The authors used myelin protein zero (MPZ) as a control of PMP22 silencing (Figure 4) providing normalized data. However, this approach is flawed, because as they show both in Figure 4 and in Figure 5 MPZ levels decrease with demyelination and increase with improved myelination, therefore the amount of PMP22 silencing is clearly overestimated (Figure 4b). The degree of PMP22 silencing should be normalized to a housekeeping gene not affected by myelination.
14. In the last paragraph of the discussion, what do the authors mean by "...measure in a less favourable situation such as at one or two months post-treatment." This is in contrast to the statement of the authors that the pathology in this model starts very early- therefore therapeutic effects could be significant already at 2 months.
15. Related to this, in Figure 7 (a-c), no statistical results are shown for functional outcomes in all the time points tested. Where they significant?
16. In the morphological analysis, have the authors looked into the degree of onion bulb formation and whether this is improved in treated animals?
17. The entire paper needs some editing for proper use of the English medical terms and language general. For example, starting from the abstract, authors should better formulate "...foot drop walking problems", "...muscle waste" should be "...muscle wasting". In the first paragraph of the introduction, correct syntax in "...Indeed, an indirect gene therapy approach consists of the transduction of muscle cells to increase their production of neurotrophin 3 in order to promote axon survival is in..." ...and many other similar errors.

Reviewers' comments:

Reviewer #1 (Remarks to the Author):

In this paper Gautier et al. evaluate the specificity and efficiency of intra-nerve injection of (AAV)-based to silence PMP22 expression using in a transgenic rat model of Charcot-Marie-Tooth disease. The authors evaluated the therapeutic efficiency of the AAV2/9 shRNAs targeting PMP22 mRNA on the human skin molecular biomarkers used to predict CMT1A disease outcome. This study shows impressive correlation results between functional measures and molecular biomarkers using a multi-varied analysis. This study shows high transduction rate of the AAV2/9 shRNAs, restricted to the injected nerves with long term benefits on rodent or NHP peripheral nervous system. This study shows very encouraging results on rodents and NHP that could be translated to human gene therapy research aiming to develop therapies for patients suffering from CMT1A, with potential application to other peripheral neuropathies.

This study is very convincing.

I have only minor points and clarifications that the authors need to address in some of their figure conclusions.

- The supporting figures and tables for the first paragraph on broad and specific transduction are missing an accurate method. It is unclear how the total mSC and transduced mSC were counted. Fig 1 a. and b., show only eGFP on cross sections, where is the mSC counterstain? Was it using CARS and Tuj1? Please clarify this in text/legend and Methods.

We acknowledge that the presentation of the data in Fig.1 was misleading. We modified Fig. 1 in order to illustrate 1-the result of the injection of AAV2/9 and 2/rh10 in the sciatic nerve of rodents (Fig. 1a and Supplementary Fig. S1, both including a negative control injected with the dye only) and 2- the nature of the GFP-positive cells seen in 1a using both myelin-detecting CARS (Fig. 1b) and immunostaining for myelin and axons (Fig. 1c). The modified text is underlined in the results section of the manuscript. The Fig. 1c also illustrates how the transduction rate presented in Table 2 was measured as detailed in the legend of Table 2 and in the Material and Methods page 22. "The percentage of transduced mSC (GFP and MBP positive cells surrounding Tuj1 positive axons) over all mSC (MBP positive but not GFP positive cells surrounding Tuj1 positive axons) in the full section was calculated using Zen software (Zeiss, France)."

- Fig.1 legend. And Table1 legend. Proximal distances from the injection site were 2, 3 and 4 cm for mice, rats and NHP respectively. Distal distances from the injection site were 0.5, 1 and 2 cm for mice, rats and NHP respectively. I have hard time to understand the nomenclature used as 'proximal' the farthest from the site of injection and 'distal' the shortest from SI?

Since the site of injection have always the highest transduction rate, the proximal transduction rate that is near the SI should be close the SI rate? and the distal rates should be higher? There is a misunderstanding on what is proximal and distal to SI, please clarify/correct.

The definition of proximal and distal regions has been illustrated in Supplementary Fig. S5a. More details on the distance values can be found in the Material and Methods page 22. "Proximally distances of the injection site were 2 cm for mice, 3 cm for rats, and 2 and 4 cm for NHP. Distally distances of the injection site were 0.5 cm for mice, 1 cm for rats and 2 cm for NHP."

- In the methods, there is no information on the injection site in the rodents while it is detailed for the NHP.

Material and Methods page 19 “AAV vectors were injected in the sciatic nerve close to the sural nerve branching proximally to the bifurcation between the common peroneal and tibial nerves (0,5 and 1 cm respectively in adult mice and rats and 0.2-0,5 cm in mouse and rat pups respectively).”

- “The transduction rate was between 21% and 69% at these distant points (Table 1). The difference resulted from the direction of the needle when inserted into the nerve.” By just reading this sentence, without looking at the table, it seems that you have a big variability on distal transduction within animal (from 21% to 69%) due to the inconsistent direction the needle was inserted. I would rephrase this sentence making a general observation for adult rodents (63% and 91%) and NPH (21% to 69%). The highest transductions rates are observed at the injection site and proximal to the SI and are getting lower distally from the SI.

We chose to rephrase as a broader result in rodent experiments because they are indeed variable “This diffusion was very significant for AAV2/9-CAG-GFP as the average transduction rate was 73% at these two distant points in both mice and rats (Table 2).” For the NHP, the diffusion was evaluated at three distinct points (two proximally and one distally of the injection site). So we chose here to indicate all the data we had “In the NHP sciatic nerve injected with AAV2/9-CAG-GFP vector, the diffusion was evaluated at three distant points distally and proximally of the injection point covering 30 to 50% of the nerve length. The transduction rate was 68% at the injection site, 69% 2 cm distally, 55% 2 cm proximally and 21% 4 cm proximally (Supplementary Table S1).”page 6.

- Figure2a. & c. In addition, typical morphological characteristics of mSC were seen through eGFP labeling such as Schmidt-Lanterman incisures (Fig. 2a, arrows), Cajal’s bands (Fig. 2a, arrowheads) and nodes of Ranvier (Fig. 2a, stars). Right panels: higher magnification showing the immunostaining for myelin MBP (red) of teased fibers. eGFP (green) is expressed in the nucleus (star), the Cajal’s bands (arrowheads) and the Schmidt-Lanterman incisures (arrows) of mSC. Some conclusions on subcellular localization of eGFP made from Fig2.a and c. are overstated regarding eGFP labelling in SLI, nodes of Ranvier, and nuclei. The authors should revised their conclusions based on the current figures or add additional evidences using appropriate co-staining for SLI, nodes of Ranvier and nuclei.

Immunostaining of the several sub-domains where GFP is localized in mSC were done and are shown in Supplementary. Fig. S2b and c except for the nucleus because it is not a particular subdomain of mSC. The results section of the manuscript has been modified accordingly. “In addition, typical morphological characteristics of mSC were seen through GFP labeling and/or subcellular markers: Schmidt-Lanterman incisures (Supplementary Fig. S2a and b, white arrows), Cajal’s bands (Supplementary Fig. S2a, blue arrows) and paranodal loops surrounding the node of Ranvier (Supplementary Fig. S2a and c, arrowheads).” Page 5.

- Figure4.a Why there is no loading control to correct quantification loading variability?

We used loading controls for WB using β -actin or whole protein as recommended by the manufacturer (<https://www.licor.com/bio/reagents/revert-total-protein-stain-for-western-blot-normalization>) (Pillai-Kastoori et al., 2020). We found that the total protein loading control was more reliable in sciatic nerve samples and in particular in diseased sciatic nerve samples than usual housekeeping genes. All these loading controls are now shown in Fig. 2 and 3 and Supplementary Fig. S6. In the experiment

shown in Fig.2 (previously Fig. 4), PMP22 (X) was normalized on MPZ (Y) expression and then this ratio was normalized on the loading control (Z)= (X/Y)/Z.

Is the variability seen in CMT1A ctrl.sh related to the variability in PMP22 overexpression in CMT1A rat model or a technical variability?

The variability seen in CMT1A ctrl.sh lines is due to the variability of PMP22 expression in nerves of CMT1A animals. More precisely, it is due to the variability of the myelination that occurs in these nerves. With the same amount of PMP22 expressed in each cell due to the gene duplication, animals but also patients produce a heterogeneous number of mSC. As PMP22 is only expressed in myelin, this results in a heterogeneous amount of PMP22. This is the reason why PMP22 expression in CMT1A nerves has to be normalized on the amount of myelin (in our work MPZ) in order to detect a significant increase due to the gene duplication.

- Figure6.b With the actual image magnification, it is hard to identify the nodes of Ranvier indicated by white arrow heads. A magnified insert is necessary to show how the node of Ranvier were identified using CARS.

A new clearer image has been added with the requested insert in Fig. 4a (previously Fig. 6b).

- Figure7a. When CMT1A animals were treated with AAV2/9-sh1 or sh2, the NCV remained not significantly different to WT ctr.sh values at all-time points for at least 12 months (Fig. 7a). That is not what the statistics on the graph are showing (**). For the NCV at least 2, 3, 9 and 12mths seem different from WT ctr.sh animals. Please correct accordingly in the text.

This is indeed an overstatement. The sentence was corrected as following “When CMT1A animals were treated with AAV2/9-sh1 or-sh2, the NCV remained close to WT ctr.sh values at all-time points for at least twelve months (Fig. 5a).” page 8. All the statistical data are presented in Supplementary Table S2.

Reviewer #2 (Remarks to the Author):

Charcot-Marie-Tooth diseases are the most common inherited disorders affecting peripheral nervous system (prevalence of 1 in 2500). Among them, CMT1A, an autosomal dominant demyelinating form of CMT caused by a uniform 1.5Mb tandem duplication on chromosome 17p accounts for 70% of demyelinating CMT cases and one third of all CMT cases and affects more than two million people worldwide. Since we have no cure for CMT1A disease, the goal to define new therapies is of an extreme importance.

Gautier and colleagues in their manuscript described a novel approach based on gene therapy as treatment for CMT1A. The animal model used for their study is the well-established line of rat for CMT1A, but also mice and non-human primates were used to test the viral-based approach. In general, the rationale of this study, the description of the methods and the objective of this research are well defined. The manuscript is also well written.

After reviewing the manuscript of Gautier and colleagues I have few points that in my opinion the authors should address:

1 – The biggest discrepancy that I noticed in this work is different efficiency of the construct “sh2” in silencing Pmp22. It seems that sh2 is able to downregulate Pmp22 in vivo (rat model) but not in vitro in rat cell line (figure 3c and also strongly stated in “Material & Methods” pag. 16 “Cloning and vector production”). Perhaps the cellular model RT4-D6P2T was not the most appropriate and primary culture of rat Schwann cells could be used for a further analysis.

We believe there is no discrepancy here but most likely a misunderstanding. Rat CMT1A model overexpresses mouse PMP22. Sh2 downregulates mouse PMP22 expression as seen in MSC80 mouse cell line but not rat PMP22 expression as seen in RT4-D6P2T rat cell line. When expressed in CMT1A rat model, sh2 downregulates overexpressed mouse PMP22 reducing the overall amount of PMP22. The PMP22 left after sh2 silencing in vivo is probably mostly rat PMP22 but we cannot show it because the rabbit anti PMP22 antibody used in Western blot experiments does not discriminate between the two species.

2 – I think that the choice of MPZ as normalizer for the experiment shown in figure 4 (a and b) is not the most appropriate. The authors should have used a glial marker (Sox10? S100b?) or an ubiquitous expressed marker (Beta-actin? Alpha-tubulin?). MPZ expression in the different animal groups is indeed an interesting data and should be maintained in this work. Also in Figure 5B I would recommend to use one of the aforementioned marker instead/in parallel to the total protein quantification.

We used loading controls for WB using β -actin or whole protein as recommended by the manufacturer (<https://www.licor.com/bio/reagents/revert-total-protein-stain-for-western-blot-normalization>) (Pillai-Kastoori et al., 2020). We found that the total protein loading control was more reliable in sciatic nerve samples and in particular in diseased sciatic nerve samples than usual housekeeping genes. All these loading controls are now shown in Fig. 2 and 3 and Supplementary Fig. S6. In the experiment shown in Fig.2 (previously Fig. 4), PMP22 (X) was normalized on MPZ (Y) expression and then this ratio was normalized on the loading control (Z) = (X/Y)/Z.

Why did we normalize PMP22 expression on myelin marker MPZ expression? As PMP22 is a protein of the myelin sheath, when myelin sheath is missing or decreased, PMP22 expression is also decreased. When PMP22 gene is duplicated, PMP22 protein expression is increased in the myelin sheath relatively to other myelin proteins (such as MPZ). However, as this duplication induces a lack or a reduction of the myelin sheath in nerves, the absolute PMP22 expression decreases in these

nerves. So, in order to measure the effect of the silencing induced by our treatment, we have to normalize PMP22 protein amount over other myelin proteins (hence MPZ) and not just on a loading control.

3 – Some of the bars used for the statistics are not clear. In detail:

- Fig 4b (CMT1A Vs sh2 is highly significant. I assume that also CMT1A Vs sh1 is the same but it is not clear from the graph.

The bars indicating the statistical comparison in the graph of Fig. 2 (previously Fig. 4) and Fig. 3 (previously Fig. 5) have been modified to clearly indicate which groups are compared.

- Fig. 7b and 7C (is wt Vs sh1 AND sh2 highly significant? Or only Wt Vs one the condition?)

This is now specified in the legend of Fig.5 (previously Fig. 7). “Statistical analysis shows two-way ANOVA followed by Tukey’s post hoc (a, b and c) comparing all groups paired two by two (all data of multiple comparison tests are available in Table S2) or one-way ANOVA followed by Tukey’s post hoc for six and twelve months post-injection in (d).” page 46.

4 – In Figure 6D a representative picture for CMT1A sh2 is needed. And if the space for the figure is sufficient also an example of CMT1A sh1 for Fig. 6A and 6B (this last 2 pictures are to be considered optional).

We added pictures illustrating the results obtained with CMT1A sh2 and CMT1A sh1 in Fig. 4b and d respectively (previously Fig. 6).

5 – Some of the biomarker with a different expression showed in Fig.8 (e.g. Nrg1-1) might reflect a preservation of axons after the viral transduction. Authors should comment this or provide evidences of lack of axonal loss in CMT1A rat model after the silencing of Pmp22.

We added the following sentence to the discussion page 16 “In addition, as these events that are linked and promoted by our treatment, it remains unclear whether the biomarkers changes reflected a higher myelin amount or a higher maintenance of axons.”

6 – Authors should defend their decision to have chosen the rat CMT1A model instead of other existing murine models, in particular relating the Pmp22 mRNA overexpression observed in human patients and the different animal models. Being Dr. Fledrich R one of the co-author it will not be difficult to address this point (I cite here 2 reviews that could be included in the discussion: “Murine therapeutic models for Charcot-Marie-Tooth (CMT) disease” BMB 2012 and ”A rat model of Charcot-Marie-Tooth disease 1A recapitulates disease variability and supplies biomarkers of axonal loss in patients” Brain, 2012).

We thank the reviewer for this judicious proposition of references. We incorporated them in our discussion of the rodent models Page 13 “While several mouse models overexpressing human PMP22 are available to mimic the disease in animals (Fledrich et al., 2012a), the rat CMT1A was chosen here because it mimics more closely the clinical aspect of the disease (Fledrich et al., 2012b). Indeed, notwithstanding the discovery of two shRNAs specifically targeting human PMP22 expression, our

goal was less to characterize a therapeutic product than to evaluate the functional benefit of such a gene therapy directly targeting the molecular and cellular causes of the disease. In fine the success of a therapy for CMT1A patients will be less based on the product itself than on the benefit (vs risk) for the patient and on the way we measure this benefit. In this regard, the validation of skin biomarkers as markers of the gene therapy efficiency in CMT1A rats constitutes an important progress.”

7 – Figures 1, 3 and 4 all the Tables might be considered as supplementary material if the format for the journal allows it.

In order to present the data more clearly and logically we recomposed the tables as Table 1 “Cellular specificity of the transduction pattern after intra-nerve injection (%)” and Table 2 “Transduction rate after intra-nerve injection (%)”. We propose to add the last table exposing the data obtained on NHP in Supplementary Table S1. Moreover, Fig. 3 was added in Supplementary Figure (Fig. S6 in the revised manuscript)

Reviewer #3 (Remarks to the Author):

CMT1A is the most common type of inherited demyelinating neuropathy resulting from duplication and overexpression of the PMP22 gene. This paper by Gautier et al., describes the development of a gene therapy approach to treat CMT1A by silencing overexpressed mouse PMP22 in a rat model of the disease using AAV mediated delivery of shRNA via direct injection into the sciatic nerve.

This is an important study as it supports the potential use of gene silencing therapy to treat CMT1A. Extensive work has been done and the results are encouraging. However, there are some major and several other issues with the study:

Major concerns:

1. There is limited potential for clinical translation of this approach with direct intraneural injections due to the invasiveness of the procedure and high demand on neurosurgical expertise.

We respectfully disagree with the reviewer concerning the degree of invasiveness of the intraneural injection of an AAV2/9 vector. Indeed, this transdermal injection does not require a surgical intervention as anaesthesiologists currently practice it for regional anaesthesia. It indeed requires a certain degree of expertise but a significant number of clinicians are trained for this practice (see KOL P. Bigeleisen's opinion in attached doc) and the injection, when performed correctly and with non toxic solution such as our vector solution, is arguably not more deleterious than perineural injection (Cappelleri et al., 2016). Indeed, while local anaesthetics are toxic for nerve fibres, solutions containing gene therapy products are not toxic. We requested opinions of several professionals and anaesthesiologists who are practicing this injection and the very large majority agreed to consider this intraneural treatment with a non-toxic product to be feasible and acceptable provided that injection is 1- interfascicular and 2- monitored through ultrasound imaging, nerve activity recording and measure of the injection pressure (see UMI survey summary and opinions of few KOLs in attached documents). All these parameters are mastered by most of the specialized anaesthesiologists. Of note, regional anaesthesia is of course also practiced on children. At least in France all major hospitals house a clinician anaesthesiologist or neurosurgeon who would be qualified to perform these injections. Finally, one the two AAV-based gene therapy treatments that have been approved for market sale, namely Luxturna, is administrated locally through an intravitreal injection (https://ec.europa.eu/health/documents/communityregister/2018/20181122142655/142655_en.pdf), which is not devoid of risk such as increased intraocular pressure, retinal tear, intraocular inflammation and/or infection related to the procedure, retinal detachment (https://www.ema.europa.eu/en/documents/rmp-summary/luxturna-epar-risk-management-plan-summary_en.pdf).

Taken together we believe that the gene therapy approach we propose, while not devoid of risks notably linked to injection route, has a clear and significant potential for clinical translation.

2. The proof-of concept is provided on silencing the mouse PMP22 by targeting the coding region. How can this be translated to treating patients with overexpression of human PMP22? Further experiments to validate the same approach for human PMP22 silencing is needed.

We thank the reviewer for this interesting proposition to increase the relevance of our work. We designed two shRNAs targeting human PMP22 in two independent region of the PMP22 mRNA and we tested them in a human cell line expressing human PMP22. In Supplementary Fig. S6, we now show that these two shRNAs significantly decrease human PMP22 expression. These data are presented in the results section of the manuscript as following “ As CMT1A results from PMP22

overexpression, we looked for small hairpin inhibitory RNAs (shRNAs) targeting human PMP22 mRNA in order to decrease PMP22 expression in CMT1A mSC. Two independent shRNAs or a control shRNA with no target were cloned in a pAAV plasmid under a U6 promoter followed by a CMV-GFP reporter cassette. Both shRNAs were found to be effective in reducing human PMP22 in HEK293 cells (Supplementary Fig. S6), showing that decreasing PMP22 expression in mSC may represent a relevant therapeutic approach for CMT1A.” page 6. The sequences of these shRNAs can be found in the Material and Methods.

3. Gene therapy in the CMT1A model was tested very early (P6-7), before the onset of the neuropathy. There is no evidence that this therapy would be beneficial after the onset of the neuropathy, which is a major issue in clinical translation. A proof of efficacy after onset in the disease models is needed.

According to the literature (Fledrich et al., 2012, 2014; Prukop et al., 2019) the onset of the neuropathy occurs right after birth in the CMT1A rat. Indeed, at the histological level, defects in peripheral nerves occur as soon as 6 days postnatal (Fledrich et al., 2012c, 2014a). Functionally, a significant reduction of the sciatic nerve conduction velocity is already observed by 30 days post natal (Fledrich et al., 2012, 2014; our data provided in Fig. 7A). As we observed that it takes 2-3 weeks for the transgene to reach full expression in mSC when transduced with a viral vector (Gonzalez et al., 2014), injecting the vector at 6-7 days postnatal constitutes a treatment at the onset of the neuropathy. According to Fledrich et al., 2014, CMT1A rat treatment with Nrg1 is more efficient at 6-18 days postnatal than from 18 to 90 days postnatal, after the neuropathy onset. That early treatment is more efficient than later treatment was confirmed in another study with dietary phospholipids in CMT1A rats (Fledrich et al., 2018). So, while there is indeed evidence that treatment at early age is more efficient than at late age, there is no evidence of the reverse. Therefore, we logically applied our gene therapy at early age and more precisely at the disease onset.

We extensively discussed this issue in the manuscript discussion page 13-14 “We treated young CMT1A animals 6 to 7 days postnatal when peripheral nerve myelination is most active because a large amount of nerve defects and of motor impairments result from the alteration of the initial phase of nerve myelination. Indeed, early nerve defects and impairments already occur in young CMT1A rats (Fledrich et al., 2014b, 2012b, 2019a). This is consistent with the disease onset occurring in the first decade in 75% of CMT1A patients (Morena et al., 2019). Moreover, the treatment of young CMT1A rats (P6 to P18) with soluble Neuregulin-1 was sufficient to halt disease progression at least until 9 weeks of age, while treatment of adult animals has only a limited impact on the disease (Fledrich et al., 2014b). We found that PMP22 silencing significantly increased MPZ protein expression in treated CMT1A nerves suggesting that myelin production is increased. Indeed, morphological analysis indicated that significantly more axons were myelinated and the myelin thickness is slightly increased (g-ratio decrease) in treated CMT1A nerves. Moreover, CARS analysis showed that internodes, the myelinated part of the axon between two nodes of Ranvier, were longer in treated CMT1A animals. As the number of myelinated segments and the length of internodes are determined early during myelination (Fernando et al., 2016), taken together this indicates that the deficit of myelination occurring early on in CMT1A rats is corrected by PMP22 silencing following our gene therapy. This is confirmed by the NCV analysis: at one month CMT1A ctr.sh rats already have a reduced NCV compared to WT ctr.sh rats due to the deficit of myelinated segment at early stages of postnatal development. The gene therapy is able to correct this defect as soon as one month, well before defects appear at the motor behavior level, indicating that this correction occurs directly on the myelination deficit at early stages. While our CARS analysis suggest that PMP22 silencing also prevents late-occurring defects such as focal hypermyelination and segmental demyelination, a benefit of the gene therapy for older diseased animals remains to be shown. Regarding a clinical application, these data suggest that treating CMT1A as early as possible, e.g in children would be more beneficial

than in adults. Treating CMT1A children in the long term using a gene therapy approach would constitute a major change as all existing pharmacological strategies target adult patients.”

4. Related to points 1 and 3, how do the authors evaluate the potential for regulatory approvals to test such invasive treatment in younger children with none or minimal manifestations?

According to the literature, it is clear today that CMT1A disease is a developmental disease that presents already in early childhood and includes histological abnormalities (Gabreëls-Festen et al., 1995), nerve conduction velocity slowing (Berciano et al., 2000; Yiu et al., 2008) and motor and sensory impairment (Burns et al., 2010; Cornett et al., 2017, 2019). As the disease is chronically evolving and does not affect the life expectancy, it is right that there are more adults that are affected than children and teenagers and these adults are often more heavily affected. However, the disease starts at early ages with defective myelination and then follows with demyelination and axonal degeneration (Fledrich et al., 2014a, 2019b). So, most of the adult patients have already experienced a significant axonal degeneration, which is consistent with the loss of the axon terminals in the skin biopsies (Manganelli et al., 2015). Assuredly, when too many axons have degenerated then correcting myelin does not constitute an optimal therapy anymore. Therefore, while we cannot exclude that preserving myelin in adult patients would not provide a benefit, it is clear that the main benefit of a gene therapy aimed to preserve myelin such as ours is addressed to young patients who have not lost too many axons already. Moreover, as gene therapy procures long term relieve compared to most of pharmacological approaches, the earlier the gene therapy treatment is applied the better for the patients.

Interestingly, the two AAV-based gene therapies validated for a commercial use now are both proposed to young children (Zolgensma, Avexis) or children and young patients before they have lost too many target cells (Luxturna, Spark Therapeutics, “Luxturna is indicated for the treatment of adult and paediatric patients with vision loss due to inherited retinal dystrophy caused by confirmed biallelic RPE65 mutations and who have sufficient viable retinal cells.”).

Finally, taken together, and after very long thoughts about this issue before we started the project, we decided to perform a proof of concept for this gene therapy targeting myelin at early ages and at the onset of the disease. This indeed constitutes in our view the best proof of concept in order to efficiently translate this gene therapy in CMT1A patients. This view is shared by the large majority of the clinical and industry professionals we consulted worldwide through an UMI survey (see the summary below) and also by specialists of regulatory affairs in Advanced Therapy Medicinal Products (Voisin Consulting Life Sciences).

References:

Berciano, J., García, A., Calleja, J., and Combarros, O. (2000). Clinico-electrophysiological correlation of extensor digitorum brevis muscle atrophy in children with Charcot–Marie–Tooth disease 1A duplication. *Neuromuscul. Disord.* 10, 419–424.

Burns, J., Ryan, M.M., and Ouvrier, R.A. (2010). Quality of life in children with Charcot-Marie-Tooth disease. *J. Child Neurol.* 25, 343–347.

Cappelleri, G., Cedrati, V.L.E., Fedele, L.L., Gemma, M., Camici, L., Loiero, M., Gallazzi, M.B., and Cornaggia, G. (2016). Effects of the Intraneural and Subparaneural Ultrasound-Guided Popliteal Sciatic Nerve Block: A Prospective, Randomized, Double-Blind Clinical and Electrophysiological Comparison. *Reg. Anesth. Pain Med.* 41, 430–437.

Cornett, K.M.D., Menezes, M.P., Shy, R.R., Moroni, I., Pagliano, E., Pareyson, D., Estilow, T., Yum, S.W., Bhandari, T., Muntoni, F., et al. (2017). Natural history of Charcot-Marie-Tooth disease during childhood. *Ann. Neurol.* 82, 353–359.

Cornett, K.M.D., Wojciechowski, E., Sman, A.D., Walker, T., Menezes, M.P., Bray, P., Halaki, M., and Burns, J. (2019). Magnetic resonance imaging of the anterior compartment of the lower leg is a biomarker for weakness, disability, and impaired gait in childhood Charcot–Marie–Tooth disease. *Muscle Nerve* 59, 213–217.

Fernando, R.N., Cotter, L., Perrin-Tricaud, C., Berthelot, J., Bartolami, S., Pereira, J.A., Gonzalez, S., Suter, U., and Tricaud, N. (2016). Optimal myelin elongation relies on YAP activation by axonal growth and inhibition by Crb3/Hippo pathway. *Nat. Commun.* 7, 12186.

Fledrich, R., Stassart, R.M., and Sereda, M.W. (2012a). Murine therapeutic models for Charcot-Marie-Tooth (CMT) disease. *Br. Med. Bull.* 102, 89–113.

Fledrich, R., Schlotter-Weigel, B., Schnizer, T.J., Wichert, S.P., Stassart, R.M., Meyer zu Hörste, G., Klink, A., Weiss, B.G., Haag, U., Walter, M.C., et al. (2012b). A rat model of Charcot–Marie–Tooth disease 1A recapitulates disease variability and supplies biomarkers of axonal loss in patients. *Brain* 135, 72–87.

Fledrich, R., Schlotter-Weigel, B., Schnizer, T.J., Wichert, S.P., Stassart, R.M., Meyer zu Hörste, G., Klink, A., Weiss, B.G., Haag, U., Walter, M.C., et al. (2012c). A rat model of Charcot–Marie–Tooth disease 1A recapitulates disease variability and supplies biomarkers of axonal loss in patients. *Brain* 135, 72.

Fledrich, R., Stassart, R.M., Klink, A., Rasch, L.M., Prukop, T., Haag, L., Czesnik, D., Kungl, T., Abdelaal, T.A.M., Keric, N., et al. (2014a). Soluble neuregulin-1 modulates disease pathogenesis in rodent models of Charcot-Marie-Tooth disease 1A. *Nat. Med.* 20, 1055–1061.

Fledrich, R., Stassart, R.M., Klink, A., Rasch, L.M., Prukop, T., Haag, L., Czesnik, D., Kungl, T., Abdelaal, T.A.M., Keric, N., et al. (2014b). Soluble neuregulin-1 modulates disease pathogenesis in rodent models of Charcot-Marie-Tooth disease 1A. *Nat. Med.* 20, 1055–1061.

Fledrich, R., Abdelaal, T., Rasch, L., Bansal, V., Schütza, V., Brügger, B., Lüchtenborg, C., Prukop, T., Stenzel, J., Rahman, R.U., et al. (2018). Targeting myelin lipid metabolism as a potential therapeutic strategy in a model of CMT1A neuropathy. *Nat. Commun.* 9.

Fledrich, R., Akkermann, D., Schütza, V., Abdelaal, T.A., Hermes, D., Schöffner, E., Soto-Bernardini, M.C., Götze, T., Klink, A., Kusch, K., et al. (2019a). NRG1 type I dependent autocrine stimulation of Schwann cells in onion bulbs of peripheral neuropathies. *Nat. Commun.* 10, 1467.

Fledrich, R., Akkermann, D., Schütza, V., Abdelaal, T.A., Hermes, D., Schöffner, E., Soto-Bernardini, M.C., Götze, T., Klink, A., Kusch, K., et al. (2019b). NRG1 type I dependent autocrine stimulation of Schwann cells in onion bulbs of peripheral neuropathies. *Nat. Commun.* 10, 1467.

Gabreëls-Festen, A.A.W.M., Bolhuis, P.A., Hoogendijk, J.E., Valentijn, L.J., Eshuis, E.J.H.M., and Gabreëls, F.J.M. (1995). Charcot-Marie-Tooth disease type 1A: morphological phenotype of the 17p duplication versus PMP22 point mutations. *Acta Neuropathol. (Berl.)* 90, 645–649.

Gonzalez, S., Fernando, R.N., Perrin-Tricaud, C., and Tricaud, N. (2014). In vivo introduction of transgenes into mouse sciatic nerve cells in situ using viral vectors. *Nat. Protoc.* 9, 1160–1169.

Manganelli, F., Nolano, M., Pisciotta, C., Provitera, V., Fabrizi, G.M., Cavallaro, T., Stancanelli, A., Caporaso, G., Shy, M.E., and Santoro, L. (2015). Charcot-Marie-Tooth disease: New insights from skin biopsy. *Neurology* 85, 1202–1208.

Morena, J., Gupta, A., and Hoyle, J.C. (2019). Charcot-Marie-Tooth: From Molecules to Therapy. *Int. J. Mol. Sci.* 20, 3419.

Pareyson, D., Saveri, P., and Pisciotta, C. (2017). New developments in Charcot–Marie–Tooth neuropathy and related diseases: *Curr. Opin. Neurol.* *30*, 471–480.

Pillai, R.S., Bhattacharyya, S.N., and Filipowicz, W. (2007). Repression of protein synthesis by miRNAs: how many mechanisms? *Trends Cell Biol.* *17*, 118–126.

Pillai-Kastoori, L., Schutz-Geschwender, A.R., and Harford, J.A. (2020). A systematic approach to quantitative Western blot analysis. *Anal. Biochem.* *593*, 113608.

Pipis, M., Rossor, A.M., Laura, M., and Reilly, M.M. (2019). Next-generation sequencing in Charcot–Marie–Tooth disease: opportunities and challenges. *Nat. Rev. Neurol.* *15*, 644–656.

Yiu, E.M., Burns, J., Ryan, M.M., and Ouvrier, R.A. (2008). Neurophysiologic abnormalities in children with Charcot-Marie-Tooth disease type 1A. *J. Peripher. Nerv. Syst.* *13*, 236–241.

Further concerns:

1. In the abstract the statement “...widespread transgene expression in myelinating Schwann cells in mouse, rat and nonhuman primate” is misleading and should be re-stated, as we only see expression in short segments of sciatic nerves and basically in no other PNS tissues. In the last paragraph of the introduction they admit that “the dispersion of the vector remained limited to the injected nerves”

We propose to modify the abstract as following: « Delivery in the sciatic nerve allowed widespread transgene expression in myelinating Schwann cells of this nerve in mouse, rat and non-human primate.”

2. In the introduction second paragraph, the prevalence for CMT1A of 5-10/10000 is too high, did the authors mean 5-10/100,000? (which would agree with the reference cited).

We thank the reviewer for detecting this error. We modified the introduction as following “The most common of these myelin-related CMT diseases is CMT1A (prevalence: 0.5-1.5/10000) (Pareyson et al., 2017; Pipis et al., 2019).” Page 3

3. At the end of page 6 it is stated that: “No downregulation beyond that of control levels was observed in CMT1A sh1 and CMT1A sh2 animals” How do the authors explain the lack of silencing also the endogenous rat Pmp22, since sh1 was effective also on the rat gene in vitro?”

Indeed, sh1 downregulates both mouse and rat PMP22 expression as seen in cell lines (Supplementary. Fig. S6). CMT1A rat mSC express both exogenous mouse and endogenous rat PMP22. So, it is likely that the expression of both species PMP22 is downregulated, but we cannot check for this as the rabbit anti PMP22 antibody does not distinguish them.

4. Related to this, a more detailed and specific investigation of mouse and rat PMP22 silencing effects should be done (for example using real time PCR with specific probes) to understand the degree of lowering expression of endogenous compared to overexpressed gene. This is also essential before discussing any HNPP-like changes resulting from excessive silencing, that can be seen only if endogenous rat PMP22 is also silenced.

We thank the reviewer for suggesting the investigation of the PMP22 mRNA regulation by the shRNAs that we used. We indeed investigated this using RT-qPCR on mRNA extracts of rat sciatic nerves. These data are presented in the results section of the manuscript page 7 “. At the mRNA level, mouse but not rat Pmp22 mRNA was upregulated relative to myelin marker Mpz in control CMT1A rats, resulting in an overall higher Pmp22 mRNA expression (Supplementary Fig. S7). However, neither AAV2/9-sh1 nor AAV2/9-sh2 treatment downregulated mouse or rat Pmp22 mRNAs expression (Supplementary Fig. S7).” In addition, we discussed these novel data as following “While we observed a decrease of PMP22 protein expression, we failed to record any decrease in the expression of mouse or rat pmp22 mRNA in treated CMT1A rats. The operating mechanism of shRNA or siRNA has been reported to be dual in cells: the interaction of some of these inhibitory molecules with the target mRNA leads or to the degradation of the target or to the block of the translation machinery (Pillai et al., 2007). Our data suggest that both sh1 and sh2 act through the second mechanism.” Page 13.

Following these new data on mRNA expression, we pushed a bit further our investigation of PMP22 expression in sciatic nerves of rat injected with control or Pmp22 shRNAs vectors. We stained sciatic nerves cryosections for PMP22 and imaged them with a confocal microscope. We found that, while transduced mSC of CMT1A ctr.sh and WT ctr.sh nerves showed a similar PMP22 expression than in non-transduced mSC, transduced mSC of CMT1A sh1 and sh2 nerves had less PMP22 than non-transduced mSC. This is shown in the new Supplementary Figure S8 and reported in the results section page 7: “Sciatic nerves sections immunostained for PMP22 also showed a decreased PMP22 protein expression in mSC of treated animals (Supplementary Fig. S8).”

5. The method of determining expression in specific cell types is not clear. What markers of non-myelinating Schwann cells were used? Did the authors look for other cell types (fibroblasts, epineurial cells?)

Transduction of mSC by AAV2/9 was extensively characterized using MBP expression (Supplementary Fig. S3a) and these cells represented an average of 91% of the transduced cells in the different conditions tested (Table 1). To characterize the 9% of transduced cells left, we first used neurofilament H and L-chain to characterize axons (Supplementary Fig. S3c, 4% in average of transduced cells, Table 1). Next, we used GFAP to characterize non myelinating Schwann cells (Supplementary Fig. S3b). The <1% of transduced cells left after that represented 1 or 2 cells every two samples and we did not look for their exact nature. Finally, these very rare uncharacterized cells were pooled with non-myelinating Schwann cells in non myelinated cells (nmc) representing 4% in average of transduced cells (Table 1).

Detail about the markers we used for axons and non-myelinating Schwann cells were added in the Material and Methods.

6. The authors should include negative control images for all their expression analysis (Figures 1 and 2). Negative control should be tissue from animals injected with only fast green dye solution (which was mixed with the vector)

Such negative controls were added in Fig. 1a, Fig. S1, Fig. S3d, Fig. S4a and Fig. S5b.

7. Please include the missing data of proximal and distal expression in rat nerves (Table 1). Where they low or not done? This is not mentioned in the text.

The measure of the transduction rate in the distal and proximal parts of the sciatic nerve was done only on rat pups and not adult. This is now specified in the legend. NA, not available (not done).

8. Please include missing data for NHP cell expression specificity (Tables 1 and 2)

This was not done on NHP nerves.

9. Please explain in the methods where exactly the quantification of expression rates was done in relation to the injection site in each species

*This is now explained in the Material and Methods. The proximal and distal location are illustrated in Supplementary. Fig. S5. “The percentage of transduced mSC (GFP and MBP positive cells surrounding *Tuj1* positive axons) over all mSC (MBP positive but not GFP positive cells surrounding *Tuj1* positive axons) in the full section was calculated using Zen software (Zeiss, France). For each species, this percentage was calculated at the injection site, proximally (toward the spinal cord) and distally (toward the paw) regarding the injection site (see Supplementary Figure S5). Proximally distances of the injection site were 2 cm for mice, 3 cm for rats, and 2 and 4 cm for NHP. Distally distances of the injection site were 0.5 cm for mice, 1 cm for rats and 2 cm for NHP.” page 22*

10. Please clarify in methods and results from where in relation to the injection side where the samples taken for VCNs determination.

In Material and Methods « We collected sciatic nerves 0.5 cm proximally of the injection site over a distance of 0.5 cm » page 26. In the legend of Table 3: « We collected 0.5 cm long sciatic nerves located 0.5 cm proximally to the injection site and the other organs. »

11. Please clarify in the methods (Vector delivery section) whether unilateral or bilateral injection into the sciatic nerve was performed for every experiment. This is confusing.

This is now specified in Material and Methods:

“For the transduction pattern analysis, pups and adult mice were unilaterally injected with 2 μ l containing 1×10^{10} vg and with 8 μ l containing 5×10^{10} vg respectively; Pups and adult rats were injected unilaterally with 1×10^{11} vg/nerve in 8 μ l and 1.8×10^{11} vg/nerve in 30 μ l respectively. Control animals were injected with sterile phosphate-buffered saline containing 0.01% Fast Green. For injection in the CMT1A rat model, rat pups were injected bilaterally with 1×10^{11} vg/nerve in 8 μ l.” page 19-20.

12. How can the result of lower tropism of AAVrh10 compared to AAV9 for NHP Schwann cells (Figure 1a) be based on a single injection of a single animal? Do the authors feel that this is a conclusive finding? Please discuss this limitation.

The use of non-human primate is strongly restricted and we were not allowed to use more animals to realize such preliminary but nevertheless important experiments. Regarding the clear results we obtained with a single animal (see Fig. S4) we are confident that our conclusion on the respective transduction efficacy of AAV2/9 and AAV2/rh10 in the sciatic nerve of NHP is right. The limitation is obvious as we clearly stated all over the manuscript that we injected only one NHP for each AAV

serotype. We notified the reason (ethic) why we could not use more NHP in the Material and Methods. As indicated in the discussion next extensive experiments should be done on NHP. However, as these experiments are long and expensive, we felt that it would be fair and professional to divulgate our NHP experiments right now.

13. The authors used myelin protein zero (MPZ) as a control of PMP22 silencing (Figure 4) providing normalized data. However, this approach is flawed, because as they show both in Figure 4 and in Figure 5 MPZ levels decrease with demyelination and increase with improved myelination, therefore the amount of PMP22 silencing is clearly overestimated (Figure 4b). The degree of PMP22 silencing should be normalized to a housekeeping gene not affected by myelination.

We used loading controls for WB using β -actin or whole protein as recommended by the manufacturer (<https://www.licor.com/bio/reagents/revert-total-protein-stain-for-western-blot-normalization>)(Pillai-Kastoori et al., 2020). We found that the total protein loading control was more reliable in sciatic nerve samples and in particular in diseased sciatic nerve samples than usual housekeeping genes. All these loading controls are now shown in Fig. 2 and 3 and Supplementary Fig. S6. In the experiment shown in Fig.2 (previously Fig. 4), PMP22 (X) was normalized on MPZ (Y) expression and then this ratio was normalized on the loading control (Z)= (X/Y)/Z.

Why did we normalize PMP22 expression on myelin marker MPZ expression? As PMP22 is a protein of the myelin sheath, when myelin sheath is missing or decreased PMP22 expression is also decreased. When PMP22 gene is duplicated, PMP22 protein expression is increased in the myelin sheath relatively to other myelin proteins (such as MPZ). However, as this duplication induces a lack or a reduction of the myelin sheath in nerves, the absolute PMP22 expression decreases in these nerves. So, in order to measure the effect of the silencing induced by our treatment, we have to normalize PMP22 protein amount over other myelin proteins (hence MPZ) and not just on a loading control.

14. In the last paragraph of the discussion, what do the authors mean by “...measure in a less favourable situation such as at one or two months post-treatment.” This is in contrast to the statement of the authors that the pathology in this model starts very early- therefore therapeutic effects could be significant already at 2 months.

CMT1A disease is a chronic disease resulting from nerve defects that accumulate over time. So, at late age (twelve months) defects are much heavier than at early age (one month). In these conditions, the phenotypic difference between a treated animal versus a non-treated animal is higher at twelve months than at one month. This should also be reflected in the biomarkers analysis. So the twelve months time-point is more favorable than the one month time-point to see a difference in the biomarkers expression. This does not mean that a significant difference cannot be seen at one month but it is just a less favorable time-point.

15. Related to this, in Figure 7 (a-c), no statistical results are shown for functional outcomes in all the time points tested. Where they significant?

Statistics shown in Fig. 5 (previously Fig. 7) are two-way ANOVA along all time-points comparing each group to each other. So a single P value is obtained for each test. However, the post test allows to analyze significance at each time point for each statistical analysis. In order to avoid the surcharge of stars in the graph, individual data for each time-point are not shown. Nevertheless, all P-values are

available in the Supplementary Table 2 and this indicated in the legend of the Figure page 46: “all data of multiple comparison tests are available in Supplementary Table S2”.

16. In the morphological analysis, have the authors looked into the degree of onion bulb formation and whether this is improved in treated animals?

No we did not analyze the amount of “onions bulbs” in CMT1A rats. “Onion bulbs” are peculiar structures appearing when several Schwann cells attempt and fail to remyelinate axons. Such structures take time to appear. As the disease occur on several dozen of years “onion bulbs” are easily seen in peripheral nerves of CMT1A patients. However, in CMT1A rats these structures are rare even after one year of disease. So, while this analysis make sense in human nerves, it is not that much relevant in CMT1A rats.

17. The entire paper needs some editing for proper use of the English medical terms and language general. For example, starting from the abstract, authors should better formulate “...foot drop walking problems”, “...muscle waste” should be “...muscle wasting”. In the first paragraph of the introduction, correct syntax in “...Indeed, an indirect gene therapy approach consists of the transduction of muscle cells to increase their production of neurotrophin 3 in order to promote axon survival is in...” ...and many other similar errors.

We thank the reviewer for his extensive review and we corrected the errors that we found.

Reviewers' Comments:

Reviewer #1:

Remarks to the Author:

The authors have done the necessary revisions and answered all the critics I raised. I have no further concerns. In my view, the authors have also sufficiently addressed the concerns raised by Reviewer 2.

I support the publication of this work.

Reviewer #4:

Remarks to the Author:

The authors addressed most of the concerns of Reviewer 3 in the resubmitted manuscript. There are, however, some minor points that still should be modified to improve the accuracy of the paper.

- Why the therapeutic approach was addressed before the onset of the neuropathy is well explained by the authors, but it should be clear in all the manuscript the results only prevented the development of the disease but not corrected it. This includes all the parameters assayed, including myelination deficits.
- Related to this, the authors suggest that for a clinical application, treating CMT1A children may be more beneficial. Since older animals have not been tested in this work, I believe this sentence is not correct.
- Since the animals were all injected unilaterally, I assume the functional data presented in Fig. 5 were obtained from ipsilateral limbs. Did they check for the data in the contralateral leg?
- I wonder how animals only treated on one leg were able to achieve such improved performance in the rotarod test, for instance. Do the authors have any explanation for that? A video of this test should be included as supplementary information.
- Related to the few number of animals that show vg in the DRG, and since there are several reports documenting AAV9 retrograde transport ability, this should be mentioned in the discussion. I was wondering if there is any correlation between the animals showing vg in peripheral tissues and those that contain neutralizing antibodies against AAVs. The number of animals tested for this assay is very limited. I believe it is risky to affirm that only 20% of the injected animals presented neutralizing factors.

Reviewer #4 (Remarks to the Author):

The authors addressed most of the concerns of Reviewer 3 in the resubmitted manuscript. There are, however, some minor points that still should be modified to improve the accuracy of the paper.

- Why the therapeutic approach was addressed before the onset of the neuropathy is well explained by the authors, but it should be clear in all the manuscript the results only prevented the development of the disease but not corrected it. This includes all the parameters assayed, including myelination deficits.

We agree that this clarification is required and we modified the manuscript accordingly

In the discussion :

« The gene therapy is able to prevent this defect as soon as one month, well before impairments appear at the motor behavior level, indicating that the benefit of the therapy occurs through an improved myelination at early stages. While our CARS analysis suggest that AAV2/9-sh1 and -sh2 treatments also prevents late-occurring defects such as focal hypermyelination and segmental demyelination, a benefit of the gene therapy for older diseased animals remains to be shown. Regarding a clinical application, these data suggest that treating CMT1A as early as possible, e.g in children, is a relevant opportunity. » Page 14

In the Results (page 7) and in the Figures legends (pages 42 and 45) we replaced « correct » by « prevent » (green underlining)

- Related to this, the authors suggest that for a clinical application, treating CMT1A children may be more beneficial. Since older animals have not been tested in this work, I believe this sentence is not correct.

We agree with the reviewer and we modified the sentence as following :

« Regarding a clinical application, these data suggest that treating CMT1A as early as possible, e.g in children, is a relevant opportunity. » Page 14

- Since the animals were all injected unilaterally, I assume the functional data presented in Fig. 5 were obtained from ipsilateral limbs. Did they check for the data in the contralateral leg?

Actually, animals were injected bilaterally. This is indicated in the manuscript pages 7, 12 (Results), 20 (Material and Methods), blue underlining

- I wonder how animals only treated on one leg were able to achieve such improved performance in the rotarod test, for instance. Do the authors have any explanation for that? A video of this test should be included as supplementary information.

Same answer as the previous point.

- Related to the few number of animals that show vg in the DRG, and since there are several reports documenting AAV9 retrograde transport ability, this should mentioned in the discussion.

We added this part (green underlining) in the following sentence in the discussion : As AAV9 are efficiently transported retrogradely when injected in muscles⁵⁵, the low amount of dorsal root ganglia and spinal cords positive for GFP mRNA in our experiments suggests that intranerve injections limits the transduction of axons.” page 14. Ref 55 : ElMallah, M. K. et al. Retrograde gene delivery to hypoglossal motoneurons using adeno-associated virus serotype 9. *Hum Gene Ther Methods* **23**, 148–156 (2012).

I was wondering if there is any correlation between the animals showing vg in peripheral tissues and those that contain neutralizing antibodies against AAVs.

Rat number which are positive for neutralizing factors	Neutralizing factor titer	Tissues tested for biodistribution	Biodistribution
178	1/500	Sciatic nerve, Drg, Lumbar spinal cord, heart, liver, spleen, kidney, brainstem	Only in Sciatic nerve and Drg (pool of L4 and L5)
157	1/500	Sciatic nerve, Drg, Heart, Liver	Only in sciatic nerve

Based on this very limited set of data, there is no correlation between a biodistribution in DRGs and the presence of neutralizing factors in the plasma.

The number of animals tested for this assay is very limited. I believe is risky to affirm that only 20% of the injected animals presented neutralizing factors.

We agree with the reviewer and we modified the sentence of the discussion simply referring to the data we show :

« *In this study, only 2 out of 10 injected animals presented neutralizing factors against AAV2/9 capsid in their blood. Moreover, the titers of these factors were low suggesting they were not abundant.* »